# FAIRNESS-AWARE MESSAGE PASSING FOR GRAPH NEURAL NETWORKS

## ABSTRACT

Graph Neural Networks (GNNs) have shown great power in various domains. However, their predictions may inherit societal biases on sensitive attributes, limiting their adoption in real-world applications. Although many efforts have been taken for fair GNNs, most existing works just adopt widely used fairness techniques in machine learning to graph domains and ignore or don't have a thorough understanding of the message passing mechanism with fairness constraints, which is a distinctive feature of GNNs. To fill the gap, we propose a novel fairness-aware message passing framework GMMD, which is derived from an optimization problem that considers both graph smoothness and representation fairness. GMMD can be intuitively interpreted as encouraging a node to aggregate representations of other nodes from different sensitive groups while subtracting representations of other nodes from the same sensitive group, resulting in fair representations. We also provide a theoretical analysis to justify that GMMD can guarantee fairness, which leads to a simpler and theory-guided variant GMMD-S. Extensive experiments on graph benchmarks show that our proposed framework can significantly improve the fairness of various backbone GNN models while maintaining high accuracy. Code and data are available at: https://anonymous.4open.science/r/GMMD-8226.

## 1 INTRODUCTION

In recent years, Graph Neural Networks (GNNs) have emerged as a powerful approach for node representation learning on graphs, facilitating various domains such as recommendation system (Fan et al., 2019; Ying et al., 2018), social network mining (Hamilton et al., 2017; Kipf & Welling, 2016) and knowledge graph (Hamaguchi et al., 2017; Wang et al., 2018). The success of GNNs relies on message passing mechanisms, which allows them to aggregate information from neighboring nodes and preserve both topological and node feature information (Kipf & Welling, 2016; Gasteiger et al., 2018; Liu et al., 2020). Despite the significant progress of GNNs, GNNs may inherit societal biases on sensitive attributes in data such as ages, genders, and races (Dai & Wang, 2021; Dong et al., 2022; Wang et al., 2022), resulting in biased predictions. Moreover, nodes with the same sensitive attributes are likely to be connected. The message passing process smooths representations of connected nodes, further segregating the representations of nodes in different sensitive groups, causing over-association of their predictions with sensitive attributes. The biased predictions largely limit the adoption of GNNs in many real-world applications, e.g., healthcare (Rajkomar et al., 2018), credit scoring (Feldman et al., 2015) and job hunting (Mehrabi et al., 2021).

Recently, many efforts have been taken to learn fair GNNs (Dai & Wang, 2021; Dong et al., 2022; Wang et al., 2022; Ling et al., 2023; Bose & Hamilton, 2019; Kose & Shen, 2022a). Most of these methods propose to adapt the power of regularization (Bose & Hamilton, 2019), adversarial debiasing (Dai & Wang, 2021; Wang et al., 2022), or contrastive learning techniques (Ling et al., 2023; Kose & Shen, 2022a) from traditional fairness models of Euclidean data to graph structured data. Additionally, some methods aim to directly remove bias from node features and graph topology (Dong et al., 2022). While these methods can alleviate bias issues, it is still hard to understand their effect in conjunction with the message passing process, which is a distinctive feature for GNN models. Training fair models in machine learning is a complex task as it involves utility and fairness performance. In the case of GNNs, the message passing process is a critical component: (i) it can greatly influence the utility performance of GNN models (Gilmer et al., 2017); and (ii) it might magnify the bias in graph (Dai & Wang, 2021). To control the trade-off between the utility and fairness of models, it is

important to design a fairness-ware message passing with a deep understanding of its mechanism about how to achieve fair prediction results. Hence, the above problems pose an important and challenging research question: *How to design a new message passing process that has high accuracy, is fairness-aware, and also provides a thorough understanding of how it can debias GNN models?*

Generally, message passing can be treated as solving an optimization problem with smoothness regularization (Ma et al., 2021b; Pan et al., 2020; Zhu et al., 2021; Chen et al., 2021; Gu et al., 2020; Fu et al., 2020). Specifically, message passing schemes are derived by the one step gradient of smoothness regularization, and they typically entail the aggregation of node features from neighboring nodes, which shows great utility performance. However, such optimization problem with smoothness regularization only considers utility without fairness. Therefore, we propose a new optimization formulation that considers both utility and fairness. By solving the new optimization problem, we obtain a novel fairness-aware message passing framework, GMMD, with utility performance ensured by the smoothness regularization, and fairness performance ensured by the fairness regularization. It can easily trade-off utility and fairness by adjusting the weight of each regularization term. Intuitively, the message passing of GMMD can be interpreted as encouraging each node $v_i$ to weightedly aggregate representations of other nodes with different sensitive attributes of $v_i$, and weightedly subtract representations of other nodes with the same sensitive attribute, which can alleviate over-association of the learned representation with sensitive attributes, resulting in fair representations with good utility ensured by smoothness. We also theoretically show that GMMD can minimize the upper bound of the metric for group fairness, resulting in fair predictions. Based on our theoretical analysis, we further propose a more efficient and simpler variant called GMMD-S.

Our main contributions are: (**i**) We propose a novel fairness-aware message passing framework GMMD to mitigate biased predictions on graph data, which guarantees graph smoothness and enhances fairness; (**ii**) We theoretically show that the proposed message passing can achieve better downstream performance with regard to fairness metrics; and (**iii**) Experimental results on real-world datasets show that GMMD exhibits a superior trade-off between prediction performance and fairness.

## 2 RELATED WORKS

**Graph Neural Networks.** Graph Neural Networks have shown great ability in representation learning on graphs (Gilmer et al., 2017; Scarselli et al., 2008). Existing GNNs can be roughly divided into spectral-based and spatial-based GNNs. Spectral-based approaches are defined according to graph theory (Kipf & Welling, 2016; Bruna et al., 2013; Defferrard et al., 2016; He et al., 2021; Yang et al., 2022; Wang & Zhang, 2022; He et al., 2022). The convolution operation is firstly proposed to graph data from spectral domain (Bruna et al., 2013). Then, a first-order approximation is used to simplify the graph convolution via GCN (Kipf & Welling, 2016). Spatial-based GNN models learn node representation by aggregating the neighbor nodes' information such as GraphSage (Hamilton et al., 2017), APPNP (Gasteiger et al., 2018) and GAT (Veličković et al., 2017). Despite their differences, most GNNs can be summarized as the message-passing framework composed of pattern extraction and interaction modeling within each layer. Recent works also show that the message passing scheme in each layer can be treated as one step gradient of solving a graph signal denoising problem (Ma et al., 2021b; Pan et al., 2020; Zhu et al., 2021; Chen et al., 2021; Gu et al., 2020; Fu et al., 2020).

**Fairness in Graph Learning.** Despite their great success, most existing GNNs just focus on the utility performance while ignoring the fairness issue, which can lead to discriminatory decisions (Dai et al., 2022). Therefore, many efforts have been taken for fair GNNs (Dai & Wang, 2021; Dong et al., 2022; Wang et al., 2022; Ling et al., 2023; Bose & Hamilton, 2019; Kose & Shen, 2022a; Rahman et al., 2019; Ma et al., 2021a; Dong et al., 2021; Li et al., 2021; Kose & Shen, 2022b). For example, adversarial debiasing is adopted to train fair GNNs by preventing an adversary from predicting sensitive attributes from learned fair node representations (Dai & Wang, 2021; Bose & Hamilton, 2019). Some recent works also study contrastive learning methods with fairness concern (Ling et al., 2023; Agarwal et al., 2021). FMP (Jiang et al., 2022) proposes a fair message passing with transparency. Even though message passing of FMP can implicitly optimize fairness constraints, it is still not effective and lacks a theoretical understanding of how its message passing process can mitigate biases in GNNs. We present a detailed discussion of these differences in Appendix A. Therefore, the previous works ignore a fairness-aware message passing component or their message passing isn't effective to debias predictions in GNNs. In this work, we develop a novel fairness-ware message passing method with a theoretical understanding of how it can give fair prediction results.

## 3   NOTATIONS AND PROBLEM DEFINITION

**Notations.** Let $\mathcal{G} = (\mathcal{V}, \mathcal{E}, \mathbf{X})$ be an undirected attributed graph, where $\mathcal{V} = \{v_1, v_2, \ldots, v_N\}$ is the set of $N$ nodes, $\mathcal{E} \subseteq \mathcal{V} \times \mathcal{V}$ is the set of edges, and $\mathbf{X}$ is the node feature matrix with the $i$-th row of $\mathbf{X}$, i.e., $\mathbf{X}_{i,:}$ as node feature of $v_i$. The graph structure can be denoted as the adjacency matrix $\mathbf{A} \in \mathbb{R}^{N \times N}$. $A_{ij} = 1$ if $(v_i, v_j) \in \mathcal{E}$; $A_{ij} = 0$ otherwise. The symmetrically normalized graph Laplacian matrix is defined as $\tilde{\mathbf{L}} = \mathbf{I} - \tilde{\mathbf{A}}$, where $\tilde{\mathbf{A}} = \hat{\mathbf{D}}^{-\frac{1}{2}} \hat{\mathbf{A}} \hat{\mathbf{D}}^{-\frac{1}{2}}$ is a normalized self-loop adjacency matrix with $\hat{\mathbf{A}} = \mathbf{A} + \mathbf{I}$ and $\tilde{\mathbf{D}}$ is the degree matrix of $\tilde{\mathbf{A}}$. $\mathcal{N}_i = \{v_j : (v_i, v_j) \in \mathcal{E}\}$ denotes the set of $v_i$ neighbors. We use $\mathbf{s} \in \{0, 1\}^N$ to denote sensitive attributes of all nodes, where the $i$-th element of $\mathbf{s}$, i.e., $s_i \in \{0, 1\}$, is the binary sensitive attribute of $v_i$. The set of nodes from group 0 ($s_i = 0$) is denoted as $\mathcal{S}_0 = \{v_i : v_i \in \mathcal{V} \wedge s_i = 0\}$. The set of nodes from group 1 ($s_i = 1$) is denoted as $\mathcal{S}_1 = \{v_i : v_i \in \mathcal{V} \wedge s_i = 1\}$. $N_1$ and $N_2$ are the number of nodes in $\mathcal{S}_0$ and $\mathcal{S}_1$.

**Graph Signal Denoising.** Recently, it is shown that many popular GNNs can be uniformly understood as solving graph signal denoising problems with various diffusion properties (Ma et al., 2021b; Pan et al., 2020; Zhu et al., 2021; Chen et al., 2021; Gu et al., 2020; Fu et al., 2020). For instance, the message passing in GCN (Kipf & Welling, 2016), GAT (Veličković et al., 2017) and APPNP (Gasteiger et al., 2018) can be considered as one gradient descent iteration for minimizing $\lambda_s \cdot \mathrm{tr}\left(\mathbf{F}^T \tilde{\mathbf{L}} \mathbf{F}\right) + \|\mathbf{F} - \mathbf{X}_{\mathrm{in}}\|_F^2$ with the initialization $\mathbf{F}^{(0)} = \mathbf{X}_{\mathrm{in}}$ and weight $\lambda_s$, where $\mathbf{X}_{\mathrm{in}}$ can be the original feature $\mathbf{X}$ or hidden features after feature transformation on $\mathbf{X}$. These hidden features can be obtained by passing $\mathbf{X}$ through several layers of MLPs. These GNN models optimize smoothness regularization over graph structure implicitly, but they may also inherit biased representations from the graph structure (Dong et al., 2022; Jiang et al., 2023).

**Problem Formulation.** For fair semi-supervised node classification, part of nodes $v_i \in \mathcal{V}_L$ are provided with labels $y_i \in \mathcal{Y}_L$, where $\mathcal{V}_L \subseteq \mathcal{V}$ is the set of labeled nodes, and $\mathcal{Y}_L$ is the corresponding label set. Given $\mathcal{G}, \mathcal{V}_L$ and $\mathbf{s}$, the goal of GNNs for fair semi-supervised node classification is to learn a function $f(\mathcal{G}) \rightarrow \mathcal{Y}_L$ and predict the labels of unlabeled nodes, where the predicted labels should maintain high accuracy whilst satisfying the fairness criteria about sensitive attributes $\mathbf{s}$.

## 4   GRAPH NEURAL NETWORK WITH FAIRNESS-AWARE MESSAGE PASSING

In this section, we introduce the proposed fairness-aware message passing framework GMMD, which aims to give fair prediction and maintain high accuracy. Unlike traditional message passing methods that optimize smoothness regularization over graphs while ignoring fairness constraints, GMMD formulates a new optimization problem that simultaneously considers smoothness and fairness. Then, by performing one-step gradient descent on this optimization function, we obtain a fairness-aware message passing approach that can learn fair and smooth representations that preserve utility and fairness performance. We also provide theoretical analysis to that GMMD can guarantee the fairness performance. Based on our theoretical analysis, a simpler and theory-guided fairness-aware message passing method (GMMD-S) is introduced with the fairness guarantee.

### 4.1   FAIRNESS-AWARE MESSAGE PASSING - GMMD

Traditional message passing methods in GNN models learn smooth representation over graphs while ignoring fairness issues, which hinders their adoption in many real-world applications (Dai & Wang, 2021; Dai et al., 2022). This motivates us to design a fairness-aware message passing method. Inspired by previous works (Ma et al., 2021b; Pan et al., 2020; Zhu et al., 2021) that treat message passing as the on step gradient of an optimization problem based on smoothness assumption of graph representations, we propose a new optimization problem with fairness considerations. Specifically, to guarantee both smoothness and fairness, a fairness-aware message passing method should be the solution of the following optimization function:

$$\min_{\mathbf{F}} h(\mathbf{F}) = h_s(\mathbf{F}) + \lambda_f h_f(\mathbf{F}) = \frac{\lambda_s}{2} \mathrm{tr}\left(\mathbf{F}^T \tilde{\mathbf{L}} \mathbf{F}\right) + \frac{1}{2} \|\mathbf{F} - \mathbf{X}_{\mathrm{in}}\|_F^2 + \lambda_f h_f(\mathbf{F}), \tag{1}$$

where $h_s(\mathbf{F}) = \frac{\lambda_s}{2} \mathrm{tr}\left(\mathbf{F}^T \tilde{\mathbf{L}} \mathbf{F}\right) + \frac{1}{2} \|\mathbf{F} - \mathbf{X}_{\mathrm{in}}\|_F^2$ is smoothness regularization to recover a clean node representation matrix $\mathbf{F} \in \mathbb{R}^{N \times d}$ and guide the smoothness of $\mathbf{F}$ over the graph. $h_f(\mathbf{F})$ is a regularization to control the fairness of $\mathbf{F}$. In this paper, we adopt Maximum Mean Discrepancy

(MMD) (Gretton et al., 2006) as fairness regularization $h_f(\mathbf{F})$ *as it enables us to gain a better understanding of the resulting message passing approach from both an intuitive and a theoretical standpoint together with great utility and fairness performance.* Note that we have a detailed discussion about motivations of using MMD in Appendix A. We leave other choices as future work. MMD is a popular estimator for measuring the distribution discrepancy of two groups. For two groups $\mathcal{S}_0$ and $\mathcal{S}_1$, MMD measures the distribution discrepancy as:

$$\ell_{\text{MMD}}(\mathbf{F}) = \frac{1}{N_0^2} \sum_{v_i \in \mathcal{S}_0} \sum_{v_j \in \mathcal{S}_0} k(\mathbf{F}_i, \mathbf{F}_j) + \frac{1}{N_1^2} \sum_{v_i \in \mathcal{S}_1} \sum_{v_j \in \mathcal{S}_1} k(\mathbf{F}_i, \mathbf{F}_j) - \frac{2}{N_0 N_1} \sum_{v_i \in \mathcal{S}_0} \sum_{v_j \in \mathcal{S}_1} k(\mathbf{F}_i, \mathbf{F}_j), \quad (2)$$

where $\mathbf{F}_i$ is the node representation of $v_i$ and $k(\mathbf{F}_i, \mathbf{F}_j)$ denotes the kernel similarity between $\mathbf{F}_i$ and $\mathbf{F}_j$. As our fairness constraint involves exactly matching the considered distributions using MMD, $k(\cdot, \cdot)$ should be characteristic kernels. As shown in (Sriperumbudur et al., 2008), RBF and Laplace are popular stationary characteristic kernels. Thus, in this paper, we set $k(\mathbf{F}_i, \mathbf{F}_j)$ to be the RBF kernel, i.e., $k(\mathbf{F}_i, \mathbf{F}_j) = \exp\left(-\alpha \|\mathbf{F}_i - \mathbf{F}_j\|^2\right)$, where $\alpha$ determines the rate at which the similarity between two data points decreases as the distance between them increases. Several works (Louizos et al., 2015; Oneto et al., 2020; Quadrianto & Sharmanska, 2017) have shown the benefits of directly using MMD as a regularizer to train fair models., e.g., we can directly adopt MMD to regularize the node representations of GNNs. However, such direct adoption doesn't consider the message passing process, which makes it challenging to comprehend the actual impact of using MMD regularization in conjunction with GNNs' message passing. In addition, we also empirically show that simple adoption can't perform well in Section 5. Hence, instead of simply using it as a regularizer, based on this new optimization problem, we adopt one-step gradient for $h(\mathbf{F})$ and propose the fairness-aware message passing based on MMD, which can be written as:

$$\mathbf{F}^{(k)} = \mathbf{F}^{(k-1)} - \gamma \nabla h\left(\mathbf{F}^{(k-1)}\right) = \mathbf{F}^{(k-1)} - \gamma(\lambda_s \tilde{\mathbf{L}} \mathbf{F}^{(k-1)} + \mathbf{F}^{(k-1)} - \mathbf{X}_{\text{in}}) + 4\gamma\lambda_f \alpha \mathbf{P} \mathbf{F}^{(k-1)}$$
$$= \left((1-\gamma)\mathbf{I} - \gamma\lambda_s \tilde{\mathbf{L}} + 4\gamma\lambda_f \alpha \mathbf{P}\right)\mathbf{F}^{(k-1)} + \gamma\mathbf{X}_{\text{in}}, \quad (3)$$

where for $i \neq j$, if $v_i$ and $v_j$ are in the same sensitive group $\mathcal{S}_t$, $P_{ij} = -k(\mathbf{F}_i^{(k-1)}, \mathbf{F}_j^{(k-1)})/N_t^2$, $t \in \{0, 1\}$. If $v_i$ and $v_j$ are in different sensitive groups, $P_{ij} = k(\mathbf{F}_i^{(k-1)}, \mathbf{F}_j^{(k-1)})/N_0 N_1$. And $P_{ii} = -\sum_{k=1, m \neq i}^{N} P_{im}$. The derivation for Eq.(3) is in Appendix B.1. Following Ma et al. (2021b), we set the learning rate $\gamma = \frac{1}{1+\lambda_s}$. Then the final *message passing of GMMD* can be written as:

$$\mathbf{F}^{(k)} = \mathbf{F}^{(k-1)} - \gamma \nabla h\left(\mathbf{F}^{(k-1)}\right) = \left((1-\gamma)(\mathbf{I} - \tilde{\mathbf{L}}) + 4\gamma\lambda_f \alpha \mathbf{P}\right)\mathbf{F}^{(k-1)} + \gamma\mathbf{X}_{\text{in}}. \quad (4)$$

The proposed message passing method can implicitly optimize both fairness and smoothness regularization in Eq.(1) to learn fair and smooth representation. To better understand this message passing mechanism in Eq.(4), we interpret it as modeling inter-sensitive-group and intra-sensitive-group relations to learn fair representation. Specifically, for two nodes $v_i$ and $v_j$ from different sensitive groups, we have $P_{ij} > 0$, meaning that $v_i$ will aggregate information from $v_j$ with the aggregation as kernel similarity $P_{ij}$; on contrary, if $v_i$ and $v_j$ are from the same sensitive group, then $P_{ij} < 0$, meaning that $v_i$ will subtract the information from $v_i$. Hence the proposed approach can mitigate bias in GNNs by aggregating representations of nodes with different sensitive attributes to bring nodes from different sensitive groups closer in the embedding space, and subtracting representations of nodes with the same sensitive attributes to prevent representations overly correlated with sensitive attributes. Therefore, our proposed message passing mechanism can mitigate the issue of biased label assignment in GNNs by modifying the way node representations are updated during message passing, promoting fairer and more accurate predictions. In order to gain a deeper comprehension of GMMD, we will also provide a theoretical understanding of our model for fairness performance.

## 4.2 THEORETICAL ANALYSIS

Next, we provide theoretical understandings of proposed GMMD. We show that GMMD can minimize the upper bound of the metric for group fairness. All detailed proofs can be found in Appendix G.

One of the most widely used fairness metrics for group fairness is demographic parity (Mehrabi et al., 2021), which ensures an equal positive rate for two groups. The difference in demographic parity, denoted as $\Delta_{\text{DP}} = |\mathbb{E}(\hat{y}|S=0) - \mathbb{E}(\hat{y}|S=1)|$, measures the fairness of the model, where $\hat{y}$ is the predicted probability for nodes being positive and $S$ is the sensitive attribute of nodes. Note that

following (Dong et al., 2022; Dai & Wang, 2021; Jiang et al., 2022) we only consider binary sensitive attributes, but our analysis can be easily extended to multivariate or continuous sensitive attributes. The details of $\Delta_{\text{DP}}$ are in Appendix F.

With the definition of $\Delta_{\text{DP}}$, we will theoretically analyze how our proposed message passing could affect the difference in demographic parity. We consider a K-layer GMMD with each layer performing fairness aware message passing as Eq.(4) and the input to the model is $\mathbf{F}^{(0)} = \mathbf{X}$. Then, we can obtain the representation $\mathbf{F}^{(K)}$ with a $K$-layer model, with both fairness and smoothness constraints Finally, a linear model is applied to $\mathbf{F}^{(K)}$ with a Softmax function to predict the probability of labels as $\hat{\mathbf{Y}} = \text{softmax}(\mathbf{F}^{(K)}\mathbf{W})$, where $\mathbf{W} \in \mathbb{R}^{d \times c}$ is the learnable weight matrix. $\hat{\mathbf{Y}} \in \mathbb{R}^{N \times c}$ is a matrix, where $i$-th row means the predicted label of node $v_i$. $d$ is the dimension of the input feature $\mathbf{X}$ and $c$ is the number of classes. We use $\mu \in \mathbb{R}^d$ to denote the sample mean of the original features of all nodes, i.e., $\mu = \sum_{v_i \in \mathcal{V}} \mathbf{X}_i/N$. Furthermore, $\mu^{(s)} \in \mathbb{R}^d$ represents the sample mean of original feature for the sensitive group $s$, $s \in \{0, 1\}$. We first demonstrate $\Delta_{\text{DP}}^{\text{Rep}}(\mathbf{F}^{(K)})$ at the representation level is the upper bound of $\Delta_{\text{DP}}$ through the following theorem:

**Theorem 4.1.** *Minimizing representation discrepancy between two sensitive groups* $\Delta_{DP}^{Rep}(\mathbf{F}^{(K)}) = \|\mathbb{E}(\mathbf{F}^{(K)} \mid S = 0) - \mathbb{E}(\mathbf{F}^{(K)} \mid S = 1)\|$ *is equivalent to minimizing* $\Delta_{DP}$ *of the model prediction for a binary classification problem, and* $\mathbf{F}^{(K)}$ *is the final output of a K layer model before the softmax function. When K is set as 2, we have:*

$$\Delta_{DP} \leq \frac{L}{2} \|\mathbf{W}\| \left( \Delta_{DP}^{Rep}(\mathbf{F}^{(K)}) + C_1 \|\mathbf{\Delta}\| \right), \tag{5}$$

*where* $C_1 = (2 + \frac{4}{N_0} + \frac{4}{N_1})^2 + 6 + \frac{8}{N_0} + \frac{8}{N_1}$ *and* $\mathbf{\Delta}$ *is the maximal deviation of* $\mathbf{X}$ *from* $\mu$ *i.e.,* $\mathbf{\Delta}_m = \max_i |\mu_m - \mathbf{X}_{i,m}|$. $L < 1$ *is the Lipschitz constant for the Softmax function.*

The proof is in Appendix G.1. Theorem 4.1 shows that minimizing $\Delta_{\text{DP}}^{\text{Rep}}$ at the representation level will minimize the fairness metric $\Delta_{\text{DP}}$. With this theorem, we can analyze the influence of our GMMD on $\Delta_{\text{DP}}^{\text{Rep}}$ to evaluate whether it promotes fairness in the classification task by reducing discrimination.

Next, we theoretically show that performing our proposed message passing to learn node representation can minimize the fairness metric at the representation level $\Delta_{\text{DP}}^{\text{Rep}}$.

**Theorem 4.2.** *For an arbitrary graph, after K-layer fairness-aware message passing with Eq.(4) on the original feature* $\mathbf{X}$ *to obtain* $\mathbf{F}^{(K)}$, *when* $K = 2$, *the consequent representation discrepancy on* $\mathbf{F}^{(K)}$ *between two sensitive groups is up bounded as:*

$$\Delta_{DP}^{Rep}(\mathbf{F}^{(K)}) \leq \left(3 - \left(\frac{1}{N_0 N_1^2} + \frac{1}{N_0^2 N_1}\right) \sum_{i \in S_0} \sum_{j \in S_1} k(\mathbf{F}_i^{(K-1)}, \mathbf{F}_j^{(K-1)})\right) C_3 + C_2, \tag{6}$$

*where* $C_2 = \|\mu^{(0)} - \mu^{(1)}\| + 8(1 + \frac{1}{N_0} + \frac{1}{N_1})^2 \|\mathbf{\Delta}\| + 2\|\mathbf{\Delta}\|$. *Also, the constant value* $C_3$ *is* $C_3 = (3 - \frac{1}{N_0} - \frac{1}{N_1})\|(\mu^{(0)} - \mu^{(1)})\| + (4 + \frac{2}{N_0} + \frac{2}{N_1})\|\mathbf{\Delta}\|$.

The proof is in Appendix G.2. Theorem 4.2 shows that $\Delta_{\text{DP}}^{\text{Rep}}(\mathbf{F}^{(K)})$ is upper bounded by the third term of the MMD loss in Eq.(2) and constant values $C_2$ and $C_3$. Therefore, combining theorem 4.2 and theorem 4.1, optimizing the third term of MMD regularization can minimize $\Delta_{\text{DP}}^{\text{Rep}}(\mathbf{F}^{(K)})$, which minimizes the upper bound of the fairness metric $\Delta_{\text{DP}}$. The proposed GMMD can implicitly optimize the MMD regularization and can be flexibly incorporated to various GNN backbones such as GAT (Veličković et al., 2017) and GIN (Xu et al., 2018) (see Appendix D). These findings highlight the efficacy of GMMD in mitigating fairness issues and facilitating various backbone GNNs. Note that theorem 4.2 can be extended to $K > 2$ and have similar results as shown in Appendix G.3.

### 4.3 Simple Fairness-aware Message Passing - GMMD-S

From the theoretical analysis in Section 4.2, we find that directly optimizing the third term of MMD regularization can also effectively minimize the upper bound of the fairness metric. This motivate us to further simplify GMMD to obtain a theory-guided method for mitigating bias for GNNs. Specifically, we simplify the fairness regularization function in Eq.(1) as $h_f(\mathbf{F}) = -\frac{1}{N_0 N_1} \sum_{v_i \in \mathcal{S}_0} \sum_{v_j \in \mathcal{S}_1} k(\mathbf{F}_i, \mathbf{F}_j)$ and obtain simplified fairness-aware message passing as follows:

$$\mathbf{F}^{(k)} = \mathbf{F}^{(k-1)} - \gamma \nabla h\left(\mathbf{F}^{(k-1)}\right) = \left((1 - \gamma)(\mathbf{I} - \tilde{\mathbf{L}}) + 4\gamma\lambda_f \alpha \widetilde{\mathbf{P}}\right)\mathbf{F}^{(k-1)} + \gamma\mathbf{X}_{\text{in}}, \tag{7}$$

where for $i \neq j$, if $v_i$ and $v_j$ are in the same sensitive group, $\widetilde{P}_{ij} = 0$. If $v_i$ and $v_j$ are in the different sensitive groups, $\widetilde{P}_{ij} = -k(\mathbf{F}_i, \mathbf{F}_j)/N_0 N_1$. And $\widetilde{P}_{ii} = -\sum_{k=1, m \neq i}^{N} \widetilde{P}_{im}, \gamma = \frac{1}{1+\lambda_s}$. The details can be found in Appendix B.2. Compared to GMMD, this simplified version doesn't need to calculate the similarity between node pairs from the same sensitive group and can greatly improve the efficiency of the algorithm. Using this message passing can immediately optimize the third term of the MMD regularization, which can provably minimize $\Delta_{\text{DP}}$ in Theorem 4.2 and Theorem 4.1. We call this simplified version as GMMD-S.

### 4.4 Model Framework and Training Method

**Model Framework.** To increase the expressive power of GMMD, we perform transformations on the original features, resulting in $\mathbf{X}_{\text{in}} = \text{MLP}_\theta(\mathbf{X})$, where $\mathbf{X}_{\text{in}} \in \mathbb{R}^{N \times c}$ and $\text{MLP}_\theta(\cdot)$ denotes $M$ layer MLP parameterized by $\theta$. Note that $c$ can be replaced with any values for hidden dimensions, but setting it as $c$ yields better empirical results in our experiment. Following (Ma et al., 2021b), we treat $\mathbf{F}^{(0)} = \mathbf{X}_{\text{in}}$. After transformations, we use fairness-aware message passing in Eq.(7) or Eq.(4) for $K$ layers, obtaining $\mathbf{F}^{(K)} = \text{GMMD}(\mathbf{A}, \mathbf{X}_{\text{in}}, \mathbf{s})$. We add Softmax function to $\mathbf{F}^{(K)}$ to get predictions. Finally, cross-entropy loss is used to train the model. Our full algorithm is given in Appendix C.

**Incorporate Different Backbones.** Moreover, our models is flexible to facilitate various GNN backbones. For the fairness-aware message passing in Eq.(4) and Eq.(7), the first term of these equations is the message passing method of GCN, $\mathbf{I} - \tilde{\mathbf{L}}$. Therefore, we can simply replace this message passing style with different GNN models. We also conduct the experiments of GAT and GIN backbones to show the flexibility of our proposed method in Section 5.4. We discuss the details of how to incorporate our model with different backbones in Appendix D.

**Efficient Training.** Calculating $\mathbf{P}$ or $\widetilde{\mathbf{P}}$ is time-consuming, where the time complexity are $\mathcal{O}(N^2 c)$ and $\mathcal{O}(N_0 N_1 c)$. To efficiently train our algorithm, we randomly sample an equal number of nodes from $\mathcal{S}_0$ and $\mathcal{S}_1$ in each training epoch to update the sampled nodes' representation by Eq.(4) or Eq.(7). Let the sampled sensitive group set be $\mathcal{S}_0'$ and $\mathcal{S}_1'$, where $|\mathcal{S}_0'|$ and $|\mathcal{S}_1'|$ are both equal to $N_s$. We can get the sampled fairness-aware message passing as follows:

$$\mathbf{F}_i^{(k)} = \mathbf{F}_i^{(k)} + 4\gamma\lambda_f\alpha\sum\nolimits_{j \in \mathcal{S}_0' \cup \mathcal{S}_1'} P_{ij}\mathbf{F}_j^{(k-1)}, \ \mathbf{F}^{(k)} = \big((1-\gamma)(\mathbf{I} - \tilde{\mathbf{L}}) + \gamma\mathbf{X}_{\text{in}}\big)\mathbf{F}^{(k-1)}, \ i \in \mathcal{S}_0' \cup \mathcal{S}_1', \ (8)$$

where we can also replace $\mathbf{P}$ with $\widetilde{\mathbf{P}}$ for our simplified fairness-aware message passing in Eq.(7). We give time complexity of our proposed method in Appendix E. A detailed training time comparison with state-of-the-art methods is given in Appendix I.4.

## 5 Experiment

In this section, we empirically evaluate the effectiveness of the proposed fairness-aware message passing GMMD on real-world graphs and analyze its behavior on graphs to gain further insights.

### 5.1 Experimental Setup

**Datasets.** We perform experiments on three widely-used datasets for fairness aware node classification, i.e., German, Credit and Bail (Agarwal et al., 2021). We provide the detailed descriptions, statistics, and sensitive attribute homophily measures of datasets in Appendix H.1.

**Baselines.** In the following part, we denote the proposed message passing method in Eq.(4) as GMMD and Eq.(7) as GMMD-S. To evaluate the effectiveness of GMMD and GMMD-S, we consider the following representative and state-of-the-art baselines for group fairness in graph domains, including NIFTY (Agarwal et al., 2021), EDITS (Dong et al., 2022), FairGNN (Dai & Wang, 2021), FMP (Jiang et al., 2022). We also adopt a baseline named MMD, which directly adds MMD regularization at the output layer of the GNN. For all baselines and our model, GCN (Kipf & Welling, 2016) is used as the backbone. For fair comparisons, we set the number of layers for GCN in all methods as 2. The details of baselines and implementations are given in Appendix H.3.

**Evaluation Protocol.** We use the default data splitting following NIFTY (Agarwal et al., 2021). We adopt AUC, F1-score and Accuracy to evaluate node classification performance. As for fairness

Table 1: Model utility and bias of node classification. The best and runner-up performance are marked with **boldface** and underline. ↑ represents the larger, the better; while ↓ represents the opposite.

| Dataset | Metric | GCN | NIFTY | EDITS | FairGNN | FairVGNN | FMP | MMD | GMMD | GMMD-S |
|---------|--------|-----|-------|-------|---------|----------|-----|-----|------|--------|
| **German** | AUC (↑) | **74.11±0.37** | 68.78±2.69 | 69.41±2.33 | 67.35±2.13 | 72.41±2.10 | 68.20±4.63 | 72.41±3.10 | 74.03±2.28 | 73.10±1.46 |
| | F1 (↑) | 82.46±0.89 | 81.40±0.54 | 81.55±0.59 | 82.01±0.26 | 82.14±0.42 | 81.38±1.30 | 81.70±0.67 | **83.13±0.64** | 82.54±0.92 |
| | ACC (↑) | **73.44±1.09** | 69.92±1.14 | 71.60±0.89 | 69.68±0.30 | 70.16±0.86 | 72.53±2.22 | 69.99±0.56 | 72.28±1.64 | 71.87±1.24 |
| | $\Delta_{DP}$ (↓) | 35.17±7.27 | 5.73±5.25 | 4.05±4.48 | 3.49±2.15 | 1.71±1.68 | 6.48±4.40 | 7.56±7.49 | 2.09±1.60 | **0.90±0.44** |
| | $\Delta_{EO}$ (↓) | 25.17±5.89 | 5.08±4.29 | 3.89±4.23 | 3.49±2.15 | 0.88±0.58 | 5.81±4.03 | 5.34±5.31 | **0.76±0.52** | 0.99±0.33 |
| **Credit** | AUC (↑) | **73.87±0.02** | 71.96±0.19 | 73.01±0.11 | 71.95±1.43 | 71.34±0.41 | 69.42±4.25 | 69.84±0.85 | 71.62±0.07 | 71.78±0.30 |
| | F1 (↑) | 81.92±0.02 | 81.72±0.05 | 81.81±0.28 | 81.84±1.19 | 87.08±0.74 | 84.99±2.78 | 85.87±0.37 | **87.57±0.01** | 86.96±0.58 |
| | ACC (↑) | 73.67±0.03 | 73.45±0.06 | 73.51±0.30 | 73.41±1.24 | 78.04±0.33 | 75.81±2.35 | 76.13±0.35 | **78.11±0.30** | 78.08±0.36 |
| | $\Delta_{DP}$ (↓) | 12.86±0.09 | 11.68±0.07 | 10.90±1.22 | 12.64±2.11 | 5.02±5.22 | 5.20±5.42 | 2.28±1.33 | **1.55±1.99** | 2.27±1.75 |
| | $\Delta_{EO}$ (↓) | 10.63±0.13 | 9.39±0.07 | 8.75±1.21 | 10.41±2.03 | 3.60±4.31 | 4.60±4.49 | 2.24±1.06 | **1.08±1.53** | 1.95±1.43 |
| **Bail** | AUC (↑) | 87.08±0.35 | 78.20±2.78 | 86.44±2.17 | 87.36±0.90 | 85.68±0.37 | 87.02±0.13 | 86.68±0.25 | 88.18±2.04 | **89.12±0.66** |
| | F1 (↑) | 79.02±0.74 | 64.76±3.91 | 75.58±3.77 | 77.50±1.69 | **79.11±0.33** | 77.46±0.40 | 77.51±0.67 | 77.92±1.99 | 78.72±0.49 |
| | ACC (↑) | 84.56±0.68 | 74.19±2.57 | 84.49±2.27 | 82.94±1.67 | 84.73±0.46 | 83.69±0.26 | 84.41±0.40 | 84.95±1.63 | **85.62±0.45** |
| | $\Delta_{DP}$ (↓) | 7.35±0.72 | 2.44±1.29 | 6.64±0.39 | 6.90±0.17 | 6.53±0.67 | 6.85±0.19 | 7.33±1.05 | 2.83±1.99 | **1.68±1.34** |
| | $\Delta_{EO}$ (↓) | 4.96±0.62 | **1.72±1.08** | 7.51±1.20 | 4.65±0.14 | 4.95±1.22 | 4.24±0.72 | 6.48±0.75 | 3.89±0.96 | 3.80±0.77 |

Figure 1: Hyper-parameter Analysis on Bail and German. A deeper color denotes a larger value.

metric, we adopt two group fairness metrics to measure the prediction bias, i.e., demographic parity $\Delta_{DP}$ and equal opportunity $\Delta_{EO}$ (Verma & Rubin, 2018), where the smaller values represent better performance. A detailed description of these two fairness metrics is shown in Appendix F.

**Setup.** We run each experiment 5 times and report the average performance. For a fair comparison, we select the best configuration of hyperparameters based on performance on the validation set for all methods. For all baselines, we use the same hyperparameter search space used in (Wang et al., 2022). The detailed setup and hyper-parameter settings are given in Appendix H.3.

## 5.2 OVERALL PERFORMANCE COMPARISON

In this subsection, we conduct experiments on real-world graphs to evaluate the utility and fairness of GMMD. Table 1 reports the average AUC, F1-score and Accuracy with standard deviation on node classification after five runs. We also report average $\Delta_{EO}$ and $\Delta_{DP}$ with standard deviation as the fairness metric. From the table, we make the following observations: (**i**) Generally, our GMMD and GMMD-S consistently attain top-tier performance, ranking either as the best or second-best across all metrics, which indicates the superiority of our model in achieving a better trade-off between model utility and fairness. (**ii**) When compared with FairGNN, FairVGNN, EDITS, NIFTY and FMP, GMMD can consistently outperform them. This improvement is mainly because our model combines fairness regularization with the original message passing process of GNNs. It can effectively learn a fair representation and achieves better model utility with smoothness regularization simultaneously. (**iii**) Our simplified model GMMD-S, which just minimizes the third term of the MMD regularization inspired from Theorem 4.2, achieves comparable or even better results compared with GMMD, which verifies our motivation to design a simpler fairness-aware message passing mechanism with less time cost to mitigate the fairness issue on GNNs by minimizing the inter-group kernel similarity.

## 5.3 PARAMETER ANALYSIS AND EMPIRICAL ANALYSIS ON THEORY

**Hyper-parameters Analysis.** Here we investigate the hyper-parameters most essential to our framework, i.e., $\lambda_f$ and $\lambda_s$ ($\gamma$ is set as $1/(1 + \lambda_s)$) to control smoothness and fairness regularization, respectively. The results are shown in Figure 1. *To better understand the influence of hyperparameters, we use homophily ratio and sensitive homophily ratio of the graphs to explain the observations*, which are discussed in Appendix H.1. From the figure, we can make the following observation: (**i**) For the

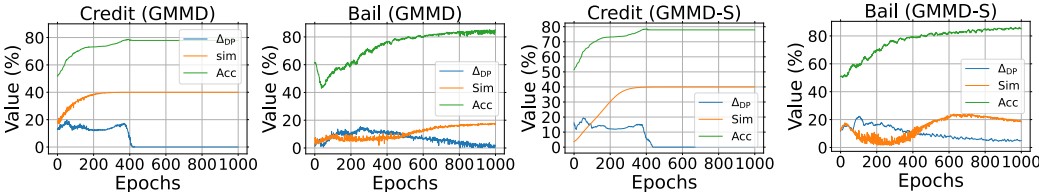

Figure 2: Empirical Analysis for the theorem on Bail and Credit datasets.

German dataset, excessively high or low weights assigned to the smoothness regularization term $\lambda_s$ lead to lower accuracy values caused by a lower homophily ratio of it, meaning that neighboring nodes may not necessarily have similar labels. For this case, it's challenging to increase the performance by simply increasing the weights of the smoothness regularization. In contrast, for the Bail dataset with a higher homophily ratio, the accuracy increases and remains stable with the increasing value of $\lambda_s$. **(ii)** Also, $\Delta_{\text{DP}}$ will drop with the increase of $\lambda_s$ for the Bail dataset with a lower sensitive homophily ratio but will enhance the bias issue for the German dataset with a higher sensitive homophily ratio. This is because encouraging similar label predictions among connected nodes via smoothness regularization can lead to consistent predictions for various groups in the Bail dataset, especially when nodes with edges are more likely to share different sensitive attributes due to lower sensitive homophily. Thus, consistent prediction results for different groups can lead to drop on $\Delta_{\text{DP}}$. However, for the German dataset with a higher sensitive homophily ratio, such regularization can exacerbate the bias by giving similar prediction results for nodes in the same sensitive group. Therefore, it's important to select $\lambda_s$ and $\lambda_f$ by considering homophily ratio and sensitive homophily ratio. **(iii)** Generally, from our experimental results, we can observe that $\lambda_s$ is better to be selected from 0.5 to 1.0 to have both utility and fairness performance guarantee. A too small (e.g., 0.1) value of $\lambda_s$ may degenerate the accuracy performance in all three datasets. Meanwhile, it's better to select $\lambda_f$ from 1 to 5 to have lower $\Delta_{\text{DP}}$. We don't include sensitivity analysis for $\alpha$, the number of MLP and GNN layers ($M$ and $K$) here, as they are not directly tied to our key motivation. Instead, we present their results in Appendix I.2.

**Empirical Analysis for Theorem 4.2.** We can understand the empirical implications of our Theorem 4.2 with Theorem 4.1 on GMMD and GMMD-S as follows: by maximizing the inter-group kernel similarity term, which is defined as $1/N_0 N_1 \sum_{i \in S_0} \sum_{j \in S_1} k(\mathbf{F}_i^{(K-1)}, \mathbf{F}_j^{(K-1)})$, we can simultaneously minimize the fairness metric. The corresponding results are shown in Figure 2, where Sim denotes the inter-group kernel similarity multiplied by 100. We can observe that $\Delta_{\text{DP}}$ will decrease with the increase of the inter-group kernel similarity. Furthermore, at the initial stages of training, the smoothness regularization will dominate the model, which will increase the accuracy performance and lead to unfair predictions. This is because the message passing of GNNs from smoothness regularization may introduce or exacerbate bias (Dong et al., 2022). Then, our proposed GMMD can minimize the inter-group similarity and help GNN models to mitigate the biased prediction results.

## 5.4 ABLATION STUDY AND DIFFERENT BACKBONES

Table 2: Ablation Study on German dataset.

**Ablation Study.** We conduct ablation studies to gain insights into the effect of message passing without smoothness and fairness regularization, which are denoted as "w/o smooth" and "w/o

| Method | German | | | | |
| --- | --- | --- | --- | --- | --- |
| | AUC (↑) | F1 (↑) | ACC (↑) | $\Delta_{\text{DP}}$ (↓) | $\Delta_{\text{eo}}$ (↓) |
| w/o smooth | 67.08±0.43 | 82.46±0.38 | 70.80±0.33 | **0.78±0.42** | 1.01±0.37 |
| w/o fair | 73.75±1.93 | **83.64±0.59** | **74.40±0.86** | 6.09±1.56 | 2.50±0.91 |
| w/o both | 69.01±6.22 | 83.06±0.86 | 73.20±1.13 | 2.92±1.50 | 3.04±1.29 |
| w/o sample | 73.68±2.18 | 83.15±0.17 | 73.37±1.00 | 1.37±0.79 | 2.21±1.51 |
| GMMD | **74.03±2.28** | 83.13±0.64 | 72.28±1.64 | 2.09±1.60 | **0.76±0.52** |

fair" in Table 5.4. We also conduct the experiment without the sampling methods, "w/o sample". First, smoothness regularization alone can achieve great performance on AUC, F1 and Acc but introduces bias and doesn't perform well on the fairness metric. Fairness regularization as an optimization target alone for GMMD produces better results on the fairness metric. The full model (last row) achieves the best performance, which illustrates that we can greatly control the trade-off between fairness and utility performance with the message passing from two regularization terms. Moreover, "w/o sample" nearly achieves the best or second-best performance across all metrics, which shows that our model with the sampling method can achieve comparable performance with the model training with all samples. It verifies the effectiveness of our sampling method to train GMMD.

**Different Backbones Analysis.** To further show the effectiveness of our model, we adopt various backbones of our model. Specifically, we replace the message passing method $(\mathbf{I} - \tilde{\mathbf{L}})$ in Eq.(4)

Table 3: Node classification performance with GIN and GAT as backbones.

| Dataset | Metric | GIN | | | | GAT | | | |
|---------|--------|---------|----------|------|--------|---------|----------|------|--------|
| | | Vanilla | FariVGNN | GMMD | GMMD-S | Vanilla | FariVGNN | GMMD | GMMD-S |
| **Bail** | AUC ($\uparrow$) | $86.14_{\pm0.25}$ | $83.22_{\pm1.60}$ | $\mathbf{86.53_{\pm1.54}}$ | $\underline{86.50_{\pm1.61}}$ | $88.10_{\pm3.52}$ | $87.46_{\pm0.69}$ | $\mathbf{90.58_{\pm0.74}}$ | $\underline{89.77_{\pm0.74}}$ |
| | F1 ($\uparrow$) | $76.49_{\pm0.57}$ | $76.36_{\pm2.20}$ | $\underline{76.62_{\pm1.88}}$ | $\mathbf{76.64_{\pm1.71}}$ | $76.80_{\pm5.54}$ | $76.12_{\pm0.92}$ | $\mathbf{78.45_{\pm3.21}}$ | $77.34_{\pm0.47}$ |
| | ACC ($\uparrow$) | $81.70_{\pm0.67}$ | $83.86_{\pm1.57}$ | $\mathbf{84.08_{\pm1.35}}$ | $\underline{84.04_{\pm1.26}}$ | $\mathbf{83.71_{\pm4.55}}$ | $81.56_{\pm0.94}$ | $83.39_{\pm4.50}$ | $83.44_{\pm2.13}$ |
| | $\Delta_{DP}$ ($\downarrow$) | $8.55_{\pm1.61}$ | $5.67_{\pm0.76}$ | $\mathbf{3.57_{\pm1.11}}$ | $\underline{3.61_{\pm1.42}}$ | $5.66_{\pm1.78}$ | $5.41_{\pm3.25}$ | $\mathbf{1.24_{\pm0.28}}$ | $3.27_{\pm1.79}$ |
| | $\Delta_{EO}$ ($\downarrow$) | $6.99_{\pm1.51}$ | $5.77_{\pm1.26}$ | $\mathbf{4.69_{\pm0.69}}$ | $\underline{4.71_{\pm0.66}}$ | $4.34_{\pm1.35}$ | $\underline{2.25_{\pm1.61}}$ | $\mathbf{1.18_{\pm0.34}}$ | $2.52_{\pm2.22}$ |

(a) Visualization on Credit  (b) Visualization on Bail  (c) German (id 413)  (d) Credit (id 12174)

Figure 3: Visualization and Case Study. For case study in (c) and (d), node color denotes the class label of nodes and node shape denotes sensitive group of nodes. Blue lines denote inter sensitive group edges and their line width is proportional to kernel similarity between the connected nodes.

and Eq.(7) with the message passing method employed in GAT and GIN. The detailed discussion of GMMD with different backbones is put into the Appendix I.1. The corresponding results on Bail dataset are given in Table 3 and results of other datasets are in Table 5 in Appendix H.3. As shown in Table 3, our GMMD and GMMD-S outperform the state-of-the-art method FairVGNN about both utility and fairness in most cases, which once again proves the effectiveness of our proposed method, and also shows that our framework is flexible to facilitate various GNN backbones.

## 5.5 VISUALIZATION AND CASE STUDY

Generally, GMMD encourages a node $v_i$ to aggregate nodes $v_j$ from a different sensitive group to mitigate fairness issues, using weights based on kernel similarity $k(\mathbf{F}_i^{(K-1)}, \mathbf{F}_j^{(K-1)})$. Next, we provide visualization and case study to understand how GMMD can preserve utility performance when learning fair representation. More results can be found in Appendix I.3. Specifically, in Figure 3 (a) and (b), we calculate the pair-wise kernel similarity of randomly sampled nodes from different sensitive groups. We visualize the distribution of similarity for these node pairs with the same labels and different labels, respectively. We can observe that node pairs having the same labels exhibit larger kernel similarity, whereas nodes with distinct labels show a lower similarity. The above observation suggests that our model assigns higher weights to nodes with the same label, thereby encouraging nodes to aggregate relevant label semantic information and preserving the utility performance. In Figure 3 (c) and (d), we randomly sample the one-hop sub-graph of a central node and calculate the kernel similarity based on learned representations between the central node and neighbor nodes. We can observe that our model can successfully assign higher weights to inter sensitive group edges which connect two nodes from the same label. This observation also demonstrates that our proposed message passing can keep utility performance by assigning higher weights to nodes with the same label from different sensitive groups.

## 6 CONCLUSION

In this paper, we propose a novel fairness-aware message passing framework GMMD to mitigate fair issues on graph data. The key idea is to derive a new message passing method for GNNs from an optimization problem with smoothness and fairness constraints. Interestingly, the derived message passing can be intuitively interpreted as encouraging a node to aggregate representations of other nodes from different sensitive groups while subtracting representation of other nodes from the same sensitive groups. By mixing node representations from various sensitive groups, GMMD can alleviate the issue of learning representations that have high correlation with sensitive attributes. Moreover, we theoretically show that GMMD enjoys good downstream performance on the fairness metric. We empirically validate our theoretical findings and our extensive experiments on several graph benchmarks validate the superiority of GMMD on both utility and fairness performance.

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

# A    ADDITIONAL RELATED WORKS

**Differences with FMP (Jiang et al., 2022)**: Our work and Jiang et al. (2022) study fairness-aware message passing from different perspectives: our model utilizes kernel-based distribution matching across various sensitive groups to mitigate bias from traditional message passing methods, while Jiang et al. (2022) considers minimizing disparities in outcomes between different sensitive groups. However, minimizing disparities can not guarantee statistical independence matching equality of opportunity (EO), while kernel-based distribution matching studied in our model can achieve this and has shown better trade-off between utility and fairness performance (Prost et al., 2019). Also, our model has achieved better utility and fairness performance from our empirical results in Table 1.

Algorithmically, our model GMMD is related to the methods of Jiang et al. (2022), but there are still important differences. FMP (Jiang et al., 2022), which uses demographic parity (DP) as the fairness regularization, mainly uses Fenchel conjugate techniques to derive message passing methods from DP due to the difficulty of directly computing the gradient of DP. In contrast, kernel-based distribution matching regularization (MMD) used in our work can be easily used to derive a fairness-aware message passing through a simple one-step gradient. Our derived message passing method has a clear intuitive understanding as discussed in section, where we can interpret our message passing as modeling inter-sensitive-group and intra-sensitive-group relations to learn fair representation. Therefore, we can understand why our message passing can mitigate bias from an intuitive standpoint.

Our theoretical analyses also have never been proposed by FMP (Jiang et al., 2022). FMP primarily emphasizes theoretical analysis on how graph topology can exacerbate bias issues in graph learning, but it lacks a theoretical guarantee regarding the effectiveness of their derived message passing in mitigating fairness issues. On the other hand, our model provides a theoretical guarantee on how it can effectively reduce the bias. This distinction underlines the theoretical strength and assurance of our model in addressing fairness concerns in GNNs.

To sum up, FMP (Jiang et al., 2022) and our model studied fairness-aware message passing from different perspectives. Kernel-based distribution matching (MMD) studied in our model can guarantee statistical independence and have better trade-off between utility and fairness performance as shown in previous works (Prost et al., 2019) and our empirical results. Furthermore, our model's message passing can be easily derived from a simple one-step gradient and has a theoretical guarantee about how our message passing can mitigate fairness issues together with an intuitive understanding. Also, we have shown great performance on both utility and fairness in experiments.

**Fair Learning with Maximum Mean Discrepancy (MMD).** In traditional fair learning, previous studies (Oneto et al., 2020; Jung et al., 2021; Prost et al., 2019; Lee et al., 2022; Louizos et al., 2015) have demonstrated that MMD exhibits better transferring ability, leading to improved performance in addressing fairness-related challenges. Additionally, MMD has shown promising results in achieving a better trade-off between accuracy and fairness (Prost et al., 2019). These findings provide a strong rationale for incorporating MMD into our approach, as it offers the potential to enhance fairness while maintaining overall model performance. Specifically, we have the following motivations or advantages of using MMD as fairness regularization in our model: (**i**) MMD regularization used in our work can be easily used to derive a fairness-aware message passing through a simple one-step gradient. Therefore, our method from MMD offers a simple format of the message passing approach, facilitating its application across various GNN architectures. (**ii**) Due to the format of our fairness aware message passing from MMD, we can provide a thorough theoretical understanding about why our message passing can help achieve a better fairness performance (as discussed in the section 4.2). (**iii**) Our derived message passing method from MMD has a better intuitive understanding as discussed in the section 4.1, where we can interpret our message passing as modeling inter-sensitive-group and intra-sensitive-group relations to learn fair representation. Therefore, we can understand why our message passing can mitigate bias issues from an intuitive standpoint.

In conclusion, MMD regularization has proven to be highly effective in enhancing the performance of traditional fair learning. This simple format of the message passing mechanism allows us to gain a comprehensive understanding of the fairness-aware message passing process based on MMD from both intuitive and theoretical perspectives. Also, our model based on MMD has promising results in experiments to resolve fairness issues on graphs.

## B OMITTED DERIVATION OF FAIRNESS AWARE MESSAGE PASSING BASED ON MMD

### B.1 DERIVATION OF FAIRNESS-AWARE MESSAGE PASSING BASED ON MMD

In this section, we provide the details of derivations of our fairness-aware message passing in Eq.(4). As shown in the Eq.(1), the optimization objective can be split into two parts $h_s(\mathbf{F})$ and $h_f(\mathbf{F})$. Following previous research (Ma et al., 2021b), we perform one-step gradient to obtain the fairness-aware message passing. The one-step gradient on $h_s(\mathbf{F})$ can be written as:

$$-\frac{\partial h_s(\mathbf{F})}{\partial \mathbf{F}} = -(\lambda_s \tilde{\mathbf{L}}\mathbf{F} + \mathbf{F} - \mathbf{X}_{\text{in}}). \tag{9}$$

Then, we use $h_f(\mathbf{F}) = l_{\text{MMD}}(\mathbf{F})$ in Eq.(2). To simplify the notation, we denote $k_{ij} = e^{-\alpha \|\mathbf{F}_i - \mathbf{F}_j\|^2}$. One-step gradient is performed on $h_s(\mathbf{F})$ to obtain:

$$-\frac{\partial h_f(\mathbf{F})}{\partial \mathbf{F}_i} = \begin{cases} \frac{4\alpha}{N_0^2}\sum_{v_j \in \mathcal{S}_0}(\mathbf{F}_i - \mathbf{F}_j)k_{ij} - \frac{4\alpha}{N_0 N_1}\sum_{v_j \in \mathcal{S}_1}(\mathbf{F}_i - \mathbf{F}_j)k_{ij}, & i \in \mathcal{S}_0, \\ \frac{4\alpha}{N_1^2}\sum_{v_j \in \mathcal{S}_1}(\mathbf{F}_i - \mathbf{F}_j)k_{ij} - \frac{4\alpha}{N_0 N_1}\sum_{v_j \in \mathcal{S}_0}(\mathbf{F}_i - \mathbf{F}_j)k_{ij}, & i \in \mathcal{S}_1. \end{cases} \tag{10}$$

This equation can be re-written as:

$$\begin{cases} (\frac{4\alpha}{N_0^2}\sum_{v_j \in \mathcal{S}_0}k_{ij} - \frac{4\alpha}{N_0 N_1}\sum_{v_j \in \mathcal{S}_1}k_{ij})\mathbf{F}_i + \frac{1}{N_0}\sum_{v_j \in \mathcal{S}_0}\mathbf{F}_j k_{ij} - \frac{1}{N_1}\sum_{v_j \in \mathcal{S}_1}\mathbf{F}_j k_{ij}, & i \in \mathcal{S}_0, \\ (\frac{4\alpha}{N_1^2}\sum_{v_j \in \mathcal{S}_1}k_{ij} - \frac{4\alpha}{N_0 N_1}\sum_{v_j \in \mathcal{S}_0}k_{ij})\mathbf{F}_i + \frac{1}{N_1}\sum_{v_j \in \mathcal{S}_1}\mathbf{F}_j k_{ij} - \frac{1}{N_0}\sum_{v_j \in \mathcal{S}_0}\mathbf{F}_j k_{ij}, & i \in \mathcal{S}_1. \end{cases}$$
$$= \sum_{j=1}^{N} 4\alpha P_{ij}\mathbf{F}_j = -\frac{\partial h_f(\mathbf{F})}{\partial \mathbf{F}_i}, \tag{11}$$

where for $i \neq j$, if $v_i$ and $v_j$ are in the same sensitive group $\mathcal{S}_0$, $P_{ij} = -k(\mathbf{F}_i^{(k-1)}, \mathbf{F}_j^{(k-1)})/N_0^2$. If $v_i$ and $v_j$ are in the same sensitive group $\mathcal{S}_1$, $P_{ij} = -k(\mathbf{F}_i^{(k-1)}, \mathbf{F}_j^{(k-1)})/N_1^2$. If $v_i$ and $v_j$ are in different sensitive groups, $P_{ij} = k(\mathbf{F}_i^{(k-1)}, \mathbf{F}_j^{(k-1)})/N_0 N_1$. And $\widetilde{P}_{ii} = -\sum_{k=1,m\neq i}^{N}\widetilde{P}_{im}$. Finally, we can get the fairness-aware message passing by performing gradient descent on the Eq.(1) with $h_f(\mathbf{F}) = l_{\text{MMD}}(\mathbf{F})$:

$$\mathbf{F}^{(k)} = \mathbf{F}^{(k-1)} - \gamma \nabla h\left(\mathbf{F}^{k-1}\right) = \mathbf{F}^{(k-1)} - \gamma \frac{\partial h_s(\mathbf{F}^{(k-1)})}{\partial \mathbf{F}^{(k-1)}} - \gamma \frac{\partial h_f(\mathbf{F}^{(k-1)})}{\partial \mathbf{F}^{(k-1)}},$$
$$= \mathbf{F}^{(k-1)} - \gamma(\lambda_s \tilde{\mathbf{L}}\mathbf{F}^{(k-1)} + \mathbf{F}^{(k-1)} - \mathbf{X}_{\text{in}}) + 4\gamma\lambda_f\alpha\mathbf{P}\mathbf{F}^{(k-1)}, \tag{12}$$
$$= ((1-\gamma)\mathbf{I} - \gamma\lambda_s\tilde{\mathbf{L}} + 4\gamma\lambda_f\alpha\mathbf{P})\mathbf{F}^{(k-1)} + \gamma\mathbf{X}_{\text{in}}.$$

### B.2 DERIVATION OF SIMPLIFIED FAIRNESS-AWARE MESSAGE PASSING

In this section, we provide the details of derivations of our simplied fairness-aware message passing in Eq.(7). Similarly, we can obtain the gradient of the smoothness function as that in Eq.(9). When $\tilde{h}_f(\mathbf{F}) = -\frac{2}{N_0 N_1}\sum_{v_i \in \mathcal{S}_0}\sum_{v_j \in \mathcal{S}_1}k\left(\mathbf{F}_i, \mathbf{F}_j\right)$. One-step gradient is performed on $\tilde{h}_f(\mathbf{F})$, which gives us:

$$-\frac{\partial \tilde{h}_f(\mathbf{F})}{\partial F_{i,m}} = \begin{cases} -\frac{4\alpha}{N_0 N_1}\sum_{v_j \in \mathcal{S}_1}(\mathbf{F}_i - \mathbf{F}_j)k_{ij}, & i \in \mathcal{S}_0, \\ -\frac{4\alpha}{N_0 N_1}\sum_{v_j \in \mathcal{S}_0}(\mathbf{F}_i - \mathbf{F}_j)k_{ij}, & i \in \mathcal{S}_1. \end{cases}$$
$$= \begin{cases} -\frac{4\alpha}{N_0 N_1}\sum_{v_j \in \mathcal{S}_1}k_{ij}\mathbf{F}_i + \frac{4\alpha}{N_0 N_1}\sum_{v_j \in \mathcal{S}_1}\mathbf{F}_j k_{ij}, & i \in \mathcal{S}_0, \\ -\frac{4\alpha}{N_0 N_1}\sum_{v_j \in \mathcal{S}_0}k_{ij}\mathbf{F}_i + \frac{4\alpha}{N_0 N_1}\sum_{v_j \in \mathcal{S}_0}\mathbf{F}_j k_{ij}, & i \in \mathcal{S}_1. \end{cases} \tag{13}$$
$$= \sum_{j=1}^{N} 4\alpha\widetilde{P}_{i,j}\mathbf{F}_j,$$

where for $i \neq j$, if $v_i$ and $v_j$ are in the same sensitive group, $\widetilde{P}_{ij} = 0$. If $v_i$ and $v_j$ are in different sensitive groups, $\widetilde{P}_{ij} = -k(\mathbf{F}_i, \mathbf{F}_j)/N_0 N_1$. And $\widetilde{P}_{ii} = -\sum_{k=1,m\neq i}^{N}\widetilde{P}_{im}$. Finally, we can get the

fairness-aware message passing by performing gradient descent on the Eq.(1) with $h_f(\mathbf{F}) = \tilde{h}_f(\mathbf{F})$:

$$
\begin{aligned}
\mathbf{F}^{(k)} &= \mathbf{F}^{(k-1)} - \gamma \nabla h\left(\mathbf{F}^{(k-1)}\right) = \mathbf{F}^{(k-1)} - \gamma \frac{\partial h_s(\mathbf{F}^{(k-1)})}{\partial \mathbf{F}^{(k-1)}} - \gamma \frac{\partial \tilde{h}_f(\mathbf{F}^{(k-1)})}{\partial \mathbf{F}^{(k-1)}}, \\
&= \mathbf{F}^{(k-1)} - \gamma(\lambda_s \tilde{\mathbf{L}} \mathbf{F}^{(k-1)} + \mathbf{F}^{(k-1)} - \mathbf{X}_{\text{in}}) + 4\gamma \lambda_f \alpha \widetilde{\mathbf{P}} \mathbf{F}^{(k-1)}, \\
&= ((1-\gamma)\mathbf{I} - \gamma \lambda_s \tilde{\mathbf{L}} + 4\gamma \lambda_f \alpha \widetilde{\mathbf{P}})\mathbf{F}^{(k-1)} + \gamma \mathbf{X}_{\text{in}}.
\end{aligned}
\tag{14}
$$

## C  THE TRAINING ALGORITHM

---
**Algorithm 1:** Training Algorithm of GMMD and GMMD-S

---
**Input:** $G = (\mathcal{V}, \mathcal{E})$
**Output:** Feature Extractor MLP with parameters $\theta$.
Initialize MLP's parameter $\theta$
**repeat**
    $\mathbf{X}_{\text{in}} \leftarrow \text{MLP}_\theta(\mathbf{X})$
    Randomly select a set nodes from $\mathcal{V}$ to obtain $\mathcal{S}_0'$ and $\mathcal{S}_1'$
    Obtain the prediction results $\hat{\mathbf{Y}}$ by Eq.(8) ($K = 2$) with the softmax function
    $\mathcal{L} \leftarrow -\sum_{v_i \in \mathcal{V}_L} \sum_{j=1}^{c} Y_{i,j} \log \hat{Y}_{i,j}$
    Update the parameters $\theta$ based on $\mathcal{L}$
**until** *convergence or reaching max iteration;*

---

The overall training algorithm is shown in Algorithm 1. We first obtain the encoded feature of nodes, $\mathbf{X}_{\text{in}}$, with $M$ layers MLP. Then, we sample a set of nodes ($\mathcal{S}_0'$ and $\mathcal{S}_1'$) for fairness regularization of GMMD or GMMD-S. We then adopt a $K$-layer GNN with each layer performing fairness-aware message passing in Eq.(8) with the softmax function. Finally, we update the parameters $\theta$ with the Cross-Entropy Loss $\mathcal{L}$ and $Y_{i,j} = \mathbb{1}(y_i = j)$ is a binary value represents whether node $i$ belongs to class $j$.

## D  DIFFERENT BACKBONES

In this section, we will illustrate our model with various backbones. As discussed in section 4.4, for the fairness-aware message passing in Eq.(4) and Eq.(7), the first term of these equations is the message passing method of GCN, $\mathbf{I} - \tilde{\mathbf{L}}$. Therefore, we can replace the first term with message passing of other GNNs. Specifically, we use GAT and GIN as examples to demonstrate how to change different message passing methods. Note that our model can be generalized to most GNN models.

### D.1  GMMD WITH GIN

Graph Isomorphism Network (GIN) adopts the message passing by the following equation (Xu et al., 2018):

$$
\mathbf{F}^{(K)} = (\mathbf{A} + (1 + \epsilon)\mathbf{I})\mathbf{F}^{(K-1)},
\tag{15}
$$

Hence, the fairness aware message passing by combining GMMD with GIN can be written as:

$$
\mathbf{F}^{(k)} = \left((1 - \gamma)(\mathbf{A} + (1 + \epsilon)\mathbf{I}) + 4\gamma \lambda_f \alpha \mathbf{P}\right)\mathbf{F}^{(k-1)} + \gamma \mathbf{X}_{\text{in}}
\tag{16}
$$

### D.2  GMMD WITH GAT

Graph Attention Networks (GAT) adopts the message passing by assigning different weights to neighbor nodes as (Veličković et al., 2017):

$$
\mathbf{F}_i^{(K)} = \sum_{i \in \mathcal{N}(i)} a_{ij}^{(K-1)} \mathbf{F}_j^{(K-1)} \quad \text{with} \quad a_{ij}^{(K-1)} = \frac{\exp\left(e_{ij}^{(K-1)}\right)}{\sum_{j \in \mathcal{N}(i)} \exp\left(e_{ij}^{(K-1)}\right)},
\tag{17}
$$

where $\mathcal{N}(i)$ is the set of neighbors of $v_i$ and $a_{ij}^{(K-1)}$ is a learnable attention score to denote the importance of node $v_j$ to $v_i$. Specifically, $a_{ij}^{(K-1)}$ is a normalized form of $e_{ij}^{(K-1)}$ for the $(K-1)$-th layer and $e_{ij}^{(K-1)}$ is calculated as:

$$e_{ij}^{(K-1)} = \text{LeakyReLU}\left(\left[\mathbf{F}_i^{(K-1)}\|\mathbf{F}_j^{(K)}\right]\mathbf{b}\right),\tag{18}$$

where $[: \| :]$ denotes the concatenation operation and $\mathbf{b} \in \mathbb{R}^{2c}$ is a learnable vector. We can denote the attention score $a_{ij}^{(K-1)}$ to a new adjaceny matrix $\mathbf{A}^{\text{att}}$, where $A_{ij}^{\text{att}} = a_{ij}^{(K-1)}$ for $(v_i, v_j) \in \mathcal{E}$; otherwise $A_{ij}^{\text{att}} = 0$. Then, we can obtain our proposed GMMD with GAT:

$$\mathbf{F}^{(k)} = \left((1-\gamma)\mathbf{A}^{\text{att}} + 4\gamma\lambda_f\alpha\mathbf{P}\right)\mathbf{F}^{(k-1)} + \gamma\mathbf{X}_{\text{in}}.\tag{19}$$

In summary, we can replace the first term of Eq.(4) and Eq.(7) with other message passing methods for most GNNs. Moreover, we can also obtain GMMD-S with GAT and GIN backbones by replacing $\mathbf{P}$ in Eq.(16) and Eq.(19) with $\widetilde{\mathbf{P}}$.

## E    EFFICIENCY ANALYSIS

### E.1    TIME COMPLEXITY ANALYSIS

We give the detailed time complexity per iteration of both GMMD and GMMD-S. Since our GMMD is agnostic to GNN models, for the time complexity analysis, we consider $K$-layer GCN as backbone, i.e., the fairness aware message passing is given in Eq.(4) and Eq.(7). A $K$-layer GCN with $c$ hidden dimensions has $\mathcal{O}\left(cK|\mathcal{E}| + Nc^2K\right)$, where $|\mathcal{E}|$ is the number of edges. For our model GMMD and GMMD-S, the extra time complexity is from our proposed fairness constrained message passing. The time complexity for the fairness constrained message passing part of GMMD and GMMD-S are $\mathcal{O}(c^2KN^2)$ and $\mathcal{O}(c^2KN_0N_1)$ with $K$ layers. The exact time complexity for these two models are $\mathcal{O}(c^2KN^2 + DK|\mathcal{E}| + Nc^2K)$ and $\mathcal{O}(c^2KN_0N_1 + cK|\mathcal{E}| + Nc^2K)$. To further improve the efficiency of our model, we randomly sample $N_s$ nodes from each sensitive group equally as discussed in the section E, the time complexity of each iteration with sampling are $\mathcal{O}(4c^2KN_s^2 + cK|\mathcal{E}| + Nc^2K)$ and $\mathcal{O}(c^2KN_s^2 + DK|\mathcal{E}| + Nc^2K)$. Note that we can also change hidden dimensions $c$ to any values.

## F    FAIRNESS METRIC FOR GROUP FAIRNESS

In this section, we discuss the group fairness metric for the binary classification problem with binary sensitive attributes. Following existing work on fair models (Mehrabi et al., 2021; Verma & Rubin, 2018), we adopt the difference in equal opportunity $\Delta_{EO}$ and demographic parity $\Delta_{DP}$ as the fairness metrics. They are defined as:

**Equal Opportunity** (Mehrabi et al., 2021): Equal opportunity requires the probability of assigning positive instances to positive class should be equal for both subgroup members, which can be written as:

$$\mathbb{E}(\hat{y} \mid S = 0, y = 1) = \mathbb{E}(\hat{y} \mid S = 1, y = 1),\tag{20}$$

where $\hat{y}$ is the predicted label and $S$ represents the sensitive attribute. The equal opportunity expects the classifier to give equal true positive rates across the subgroups. In this paper, we report the difference for equal opportunity ($\Delta_{\text{EO}}$) (Madras et al., 2018), which is defined as:

$$\Delta_{\text{EO}} = |\mathbb{E}(\hat{y} \mid S = 0, y = 1) - \mathbb{E}(\hat{y} \mid S = 1, y = 1)|.\tag{21}$$

**Demographic Parity** (Mehrabi et al., 2021): Demographic Parity requires the predictions to be independent with the sensitive attribute $S$, where $\hat{y} \perp S$. It can be formally written as:

$$\mathbb{E}(\hat{y} \mid S = 0) = \mathbb{E}(\hat{y} \mid S = 1).\tag{22}$$

The difference in Demographic Parity (Madras et al., 2018) is defined as :

$$\Delta_{\text{DP}} = |\mathbb{E}(\hat{y} \mid S = 0) - \mathbb{E}(\hat{y} \mid S = 1)|\tag{23}$$

Note that the difference between equal opportunity and demographic parity measures fairness from different dimensions. Difference of equal opportunity needs similar performance across sensitive groups; while demographic parity focuses more on fair demographics. *The smaller $\Delta_{EO}$ and $\Delta_{DP}$ are, the more fair the model is.*

# G    PROOF OF THEOREMS

## G.1    PROOF OF THEOREM 4.1

**Theorem 4.1.** *Minimizing representation discrepancy between two sensitive groups $\Delta_{DP}^{Rep}(\mathbf{F}^{(K)}) = \|\mathbb{E}(\mathbf{F}^{(K)} \mid S = 0) - \mathbb{E}(\mathbf{F}^{(K)} \mid S = 1)\|$ is equivalent to minimizing $\Delta_{DP}$ of the model prediction for a binary classification problem, and $\mathbf{F}^{(K)}$ is the final output of a $K$ layer model before the softmax function. When $K$ is set as 2, we have:*

$$\Delta_{DP} \leq \frac{L}{2}\|\mathbf{W}\| \left(\Delta_{DP}^{Rep}(\mathbf{F}^{(K)}) + C_1\|\mathbf{\Delta}\|\right), \tag{24}$$

*where $C_1 = (2 + \frac{4}{N_0} + \frac{4}{N_1})^2 + 6 + \frac{8}{N_0} + \frac{8}{N_1}$ and $\mathbf{\Delta}$ is the maximal deviation of $\mathbf{X}$ from $\mu$ i.e., $\mathbf{\Delta}_m = \max_i |\mu_m - \mathbf{X}_{i,m}|$. $L < 1$ is the Lipschitz constant for the Softmax function.*

*Proof.* To simplify the notation, we use $g(*)$ to replace the Softmax function $\mathrm{Softmax}(*)$ in the following part. $c$ is the number of the classes and we consider $c$ as 2 following current papers on group fairness. For the binary classification problem, the demographic parity distance $\Delta_{DP}$ is equivalent to the absolute expected difference in classifier outcomes between the two groups $\Delta_{DP} = \left|\frac{1}{N_0}\sum_{i \in \mathcal{S}_0} \hat{y}_{i,1} - \frac{1}{N_1}\sum_{j \in \mathcal{S}_1} \hat{y}_{j,1}\right|$ (Madras et al., 2018), where $\hat{\mathbf{y}}_j = [\hat{y}_{j,0}, \hat{y}_{j,1}]^T$ and $\hat{y}_{j,1}$ denotes the predicted probability of being positive for node $v_j$. $D$ is the dimension of $\mathbf{F}^{(K)}$. Before discussing the derivation of our theorem, we will introduce some properties of the Softmax function for the binary classification function. Considering the Softmax function $g(*)$ on two vectors with two dimensions, $\mathbf{x} \in \mathbb{R}^2$ and $\mathbf{y} \in \mathbb{R}^2$, the output of the Softmax function on these two vectors can be written as $\mathbf{x} \in [1 - a_1, a_1]$ and $\mathbf{y} \in [1 - b_1, b_1]$. $\mathrm{Softmax}(*)$ is Lipschitz continuous with Lipschitz constant $L$, i.e., $\|g(\mathbf{x}) - g(\mathbf{y})\| \leq L\|\mathbf{x} - \mathbf{y}\|$. We can have the following equation:

$$\begin{aligned}\|g(\mathbf{x}) - g(\mathbf{y})\| &= |a_1 - b_1| + |1 - a_1 - (1 - b_1)| = 2|a_1 - b_1| \\ &= 2|g(\mathbf{x})_1 - g(\mathbf{y})_1| \leq L\|\mathbf{x} - \mathbf{y}\|,\end{aligned} \tag{25}$$

where $g(\mathbf{x})_1$ represents the value of the dimension 1 for the output of $g(\mathbf{x})_1$. Based on these properties, we can get the equations for $\Delta_{DP}$ and $\Delta_{DP}^{\mathrm{Rep}}$:

$$\begin{aligned}\Delta_{DP} &= \left|\frac{1}{N_0}\sum_{i \in \mathcal{S}_0} g(\mathbf{F}_i^{(K)}\mathbf{W})_1 - \frac{1}{N_1}\sum_{j \in \mathcal{S}_1} g(\mathbf{F}_j^{(K)}\mathbf{W})_1\right|, \\ \Delta_{DP}^{\mathrm{Rep}} &= \left\|\frac{1}{N_0}\sum_{i \in \mathcal{S}_0} \mathbf{F}_i^{(K)} - \frac{1}{N_1}\sum_{j \in \mathcal{S}_1} \mathbf{F}_j^{(K)}\right\| = \left\|\mu_K^{(0)} - \mu_K^{(1)}\right\|.\end{aligned} \tag{26}$$

For a node $v_i \in \mathcal{S}_0$, we can write $\mathbf{F}_i^{(K)} = \mu_K^{(0)} + \sigma_i^K$, where $\mu_K^{(0)}$ is the mean of node representation from group 0, and $\sigma_i^K$ represents the difference between the mean representation and the representation of node $v_i$. Similarly, for a node $v_i \in \mathcal{S}_1$, $\mathbf{F}_i^{(K)} = \mu_K^{(1)} + \sigma_i^K$, where $\mu_K^{(1)}$ is the mean of node representation from group 1. From Eq.(25) and using node $v_i$ from the group 0 as an example, we can get:

$$2\left|g(\mathbf{F}_i^{(K)}\mathbf{W})_1 - g(\mu_K^{(0)}\mathbf{W})_1\right| \leq L\|\sigma_i^K\mathbf{W}\| \tag{27}$$

This gives us

$$g(\mu_K^{(0)}\mathbf{W})_1 - \frac{L}{2}\|\sigma_i^K\mathbf{W}\| \leq g((\mu_K^{(0)} + \sigma_{K,i})\mathbf{W})_1 = g(\mathbf{F}_i^{(K)}\mathbf{W})_1 \leq g(\mu_K^{(0)}\mathbf{W})_1 + \frac{L}{2}\|\sigma_i^K\mathbf{W}\|. \tag{28}$$

Based on Equations ([25](#)), ([28](#)), we have the following inequality holds:

$$g(\mu_K^{(0)}\mathbf{W})_1 - g(\mu_K^{(1)}\mathbf{W})_1 - \frac{1}{N_0}\sum_{i=1}^{N_0}\frac{L}{2}\|\sigma_i^K\mathbf{W}\| - \frac{1}{N_1}\sum_{j=1}^{N_1}\frac{L}{2}\|\sigma_j^K\mathbf{W}\|$$

$$\leq \frac{1}{N_0}\sum_{i\in\mathcal{S}_0}g(\mathbf{F}_i^{(K)}\mathbf{W})_1 - \frac{1}{N_1}\sum_{j\in\mathcal{S}_1}g(\mathbf{F}_j^{(K)}\mathbf{W})_1 \leq \tag{29}$$

$$g(\mu_K^{(0)}\mathbf{W})_1 - g(\mu_K^{(1)}\mathbf{W})_1 + \frac{1}{N_0}\sum_{i=1}^{N_0}\frac{L}{2}\|\sigma_i^K\mathbf{W}\| + \frac{1}{N_1}\sum_{j=1}^{N_1}\frac{L}{2}\|\sigma_i^K\mathbf{W}\|$$

Define $a = g(\mu_K^{(0)}\mathbf{W})_1 - g(\mu_K^{(1)}\mathbf{W})_1 - \frac{1}{N_0}\sum_{i=1}^{N_0}\frac{L}{2}\|\sigma_i^K\mathbf{W}\| - \frac{1}{N_1}\sum_{j=1}^{N_1}\frac{L}{2}\|\sigma_j^K\mathbf{W}\|$ and $b = g(\mu_K^{(0)}\mathbf{W})_1 - g(\mu_K^{(1)}\mathbf{W})_1 + \frac{1}{N_0}\sum_{i=1}^{N_0}\frac{L}{2}\|\sigma_i^K\mathbf{W}\| + \frac{1}{N_1}\sum_{j=1}^{N_1}\frac{L}{2}\|\sigma_i^K\mathbf{W}\|$. Eq.([29](#)) leads to:

$$\Delta_{\text{DP}} = |\frac{1}{N_0}\sum_{i\in\mathcal{S}_0}g(\mathbf{F}_i^{(K)}\mathbf{W})_1 - \frac{1}{N_1}\sum_{j\in\mathcal{S}_1}g(\mathbf{F}_j^{(K)}\mathbf{W})_1| \leq \max(|a|,|b|). \tag{30}$$

If we consider the case, $|a| \geq |b|$:

$$\Delta_{\text{DP}} \leq \left|g(\mu_K^{(0)}\mathbf{W})_1 - g(\mu_K^{(1)}\mathbf{W})_1 - \frac{1}{N_0}\sum_{i=1}^{N_0}\frac{L}{2}\|\sigma_i^K\mathbf{W}\| - \frac{1}{N_1}\sum_{j=1}^{N_1}\frac{L}{2}\|\sigma_j^K\mathbf{W}\|\right|$$

$$\leq \left|g(\mu_K^{(0)}\mathbf{W})_1 - g(\mu_K^{(1)}\mathbf{W})_1\right| + \frac{1}{N_0}\sum_{i=1}^{N_0}\frac{L}{2}\|\sigma_i^K\mathbf{W}\| + \frac{1}{N_1}\sum_{j=1}^{N_1}\frac{L}{2}\|\sigma_j^K\mathbf{W}\|$$

$$\leq \left|g(\mu_K^{(0)}\mathbf{W})_1 - g(\mu_K^{(1)}\mathbf{W})_1\right| + \frac{1}{N_0}\sum_{i=1}^{N_0}\frac{L}{2}\|\sigma_i^K\|\|\mathbf{W}\| + \frac{1}{N_1}\sum_{j=1}^{N_1}\frac{L}{2}\|\sigma_j^K\|\|\mathbf{W}\| \tag{31}$$

$$\leq \left|g(\mu_K^{(0)}\mathbf{W})_1 - g(\mu_K^{(1)}\mathbf{W})_1\right| + \frac{L}{2}\|\mathbf{W}\|\|\mathbf{\Delta}^K\| + \frac{L}{2}\|\mathbf{W}\|\|\mathbf{\Delta}^K\|,$$

where $\mathbf{\Delta}_m^K = \max_i|\sigma_{i,m}^K|$ and $\sigma_{i,m}^K$ represents the $m$-th dimension of $\sigma_{i,m}^K$. Similarly, we can get the same inequality for the case $|a| < |b|$. Based on Eq.([25](#)), we can get:

$$\Delta_{\text{DP}} \leq \left|g(\mu_K^{(0)}\mathbf{W})_1 - g(\mu_K^{(1)}\mathbf{W})_1\right| + \frac{L}{2}\|\mathbf{W}\|(\|\mathbf{\Delta}^K\|) + \frac{L}{2}\|\mathbf{W}\|(\|\mathbf{\Delta}^K\|)$$

$$\leq \frac{L}{2}\left\|\mu_K^{(0)}\mathbf{W} - \mu_K^{(1)}\mathbf{W}\right\| + \frac{L}{2}\|\mathbf{W}\|\|\mathbf{\Delta}^K\| + \frac{L}{2}\|\mathbf{W}\|\|\mathbf{\Delta}^K\| \tag{32}$$

$$\leq \frac{L}{2}\|\mathbf{W}\|\left\|\mu_K^{(0)} - \mu_K^{(1)}\right\| + \frac{L}{2}\|\mathbf{W}\|\|\mathbf{\Delta}^K\| + \frac{L}{2}\|\mathbf{W}\|\|\mathbf{\Delta}^K\|$$

$$= \frac{L}{2}\|\mathbf{W}\|\left(\Delta_{\text{DP}}^{\text{Rep}} + 2\|\mathbf{\Delta}^K\|\right).$$

Then, we can build the relation between $\|\mathbf{\Delta}^K\|$ and $\|\mathbf{\Delta}^{K-1}\|$. We first introduce some inequalities. For each node $v_i$, we have $\mu_m^K - \max_i|\sigma_{i,m}^K| \leq F_{i,m}^{(K)} \leq \mu_m^K + \max_i|\sigma_{i,m}^K|$, where $\mu_m^K = \frac{1}{N}\sum_{i=1}^N F_{i,m}^{(K)}$ and $\max_i|\sigma_{i,m}^K| = |\Delta_m^K|$. To simplify notation, we write it as $F_{i,m}^{(K)} \in [\mu_m^K \pm |\Delta_m|]$ For the node representation $F_{i,m}^K$ based on Eq.([4](#)) and the previous inequalities, we have the following inequality:

$$F_{i,m}^{(K)} = X_{q,m} + \frac{1}{|\mathcal{N}_i|}\sum_{v_q\in\mathcal{N}_i}F_{q,m}^{(K-1)} - \frac{1}{N_0^2}\sum_{q\in S_0}k(\mathbf{F}_q^{(K-1)},\mathbf{F}_i^{(K-1)})F_{q,m}^{(K-1)} + \frac{1}{N_0 N_1}\sum_{q\in S_1}k(\mathbf{F}_q^{(K-1)},\mathbf{F}_i^{(K-1)})F_{q,m}^{(K-1)}$$

$$+ \left(\frac{1}{N_0^2}\sum_{q\in S_0}k(\mathbf{F}_q^{(K-1)},\mathbf{F}_i^{(K-1)}) - \frac{1}{N_0 N_1}\sum_{q\in S_1}k(\mathbf{F}_q^{(K-1)},\mathbf{F}_i^{(K-1)})\right)F_{q,m}^{(K-1)}$$

$$\in \left[\mu_m^K + \mu_m^0 \pm (1 + \frac{2}{N_0} + \frac{2}{N_1})|\Delta_m^{K-1}| + |\Delta_m^0|\right], \tag{33}$$

where $\mathcal{N}_i$ represents the neighbors set of the node $v_i$. Then, we can get the inequality for $\sigma_{i,m}^K$:

$$\sigma_{i,m}^K = F_{i,m}^{(K)} - \frac{1}{N}\sum_{j=1}^N F_{j,m}^{(K)} \in \left[\pm(2 + \frac{4}{N_0} + \frac{4}{N_0})|\Delta_m^{K-1}| + 2|\Delta_m^0|\right],$$

$$\Delta_m^K = \max_i |\sigma_{i,m}^K| \le (2 + \frac{4}{N_0} + \frac{4}{N_1})|\Delta_m^{K-1}| + 2|\Delta_m^0|. \tag{34}$$

We finish building the relation between $\|\boldsymbol{\Delta}^K\|$ and $\|\boldsymbol{\Delta}^{K-1}\|$:

$$\|\boldsymbol{\Delta}^K\| \le (2 + \frac{4}{N_0} + \frac{4}{N_1})\|\boldsymbol{\Delta}^{K-1}\| + 2\|\boldsymbol{\Delta}^0\|. \tag{35}$$

In this paper, we consider the two layer GNN, where $K = 2$. We can get:

$$\|\boldsymbol{\Delta}^2\| \le \left((2 + \frac{4}{N_0} + \frac{4}{N_1})^2 + 6 + \frac{8}{N_0} + \frac{8}{N_1}\right)\|\boldsymbol{\Delta}^0\|. \tag{36}$$

Finally, from Eq.(32) and the previous equation, we have:

$$\Delta_{\text{DP}} \le \frac{L}{2}\|\mathbf{W}\|\left(\Delta_{\text{DP}}^{\text{Rep}} + C_1\|\boldsymbol{\Delta}\|\right), \tag{37}$$

where $C_1 = (2 + \frac{4}{N_0} + \frac{4}{N_1})^2 + 6 + \frac{8}{N_0} + \frac{8}{N_1}$ and we use $\|\boldsymbol{\Delta}\|$ to represent $\|\boldsymbol{\Delta}^K\|$ for $K = 0$. This completes our proof. □

### G.2 Proof of Theorem 4.2

**Theorem 4.2.** *For an arbitrary graph, after $K$-layer fairness-aware message passing with Eq.(4) on the original feature $\mathbf{X}$ to obtain $\mathbf{F}^{(K)}$, when $K = 2$, the consequent representation discrepancy on $\mathbf{F}^{(K)}$ between two sensitive groups is up bounded as:*

$$\Delta_{DP}^{Rep}(\mathbf{F}^{(K)}) \le \left(3 - (\frac{1}{N_0 N_1^2} + \frac{1}{N_0^2 N_1})\sum_{i\in S_0}\sum_{j\in S_1} k(\mathbf{F}_i^{(K-1)}, \mathbf{F}_j^{(K-1)})\right)C_3 + C_2, \tag{38}$$

*where $C_2 = \|\mu^{(0)} - \mu^{(1)}\| + 8(1 + \frac{1}{N_0} + \frac{1}{N_1})^2\|\boldsymbol{\Delta}\| + 2\|\boldsymbol{\Delta}\|$. Also, the constant value $C_3$ is $C_3 = (3 - \frac{1}{N_0} - \frac{1}{N_1})\|(\mu^{(0)} - \mu^{(1)})\| + (4 + \frac{2}{N_0} + \frac{2}{N_1})\|\boldsymbol{\Delta}\|$.*

*Proof.* $\Delta_{\text{DP}}^{\text{Rep}}(\mathbf{F}^{(K)}) = \left\|\frac{1}{N_0}\sum_{i\in\mathcal{S}_0}\mathbf{F}_i^{(K)} - \frac{1}{N_1}\sum_{j\in\mathcal{S}_1}\mathbf{F}_j^{(K)}\right\|$. The feature representation of conducting one fair-aggregation for $v \in S_0$ is based on the Eq.(4):

$$\mathbf{F}_i^{(K)} = \mathbf{X}_i + \frac{1}{|\mathcal{N}_i|}\sum_{v_u\in\mathcal{N}_u}\mathbf{F}_u^{(K-1)} - \frac{1}{N_0^2}\sum_{u\in S_0} k(\mathbf{F}_u^{(K-1)}, \mathbf{F}_i^{(K-1)})\mathbf{F}_u^{(K-1)} + \frac{1}{N_0 N_1}\sum_{u\in S_1} k(\mathbf{F}_u^{(K-1)}, \mathbf{F}_i^{(K-1)})\mathbf{F}_u^{(K-1)}$$

$$+ (\frac{1}{N_0^2}\sum_{u\in S_0} k(\mathbf{F}_u^{(K-1)}, \mathbf{F}_i^{(K-1)}) - \frac{1}{N_0 N_1}\sum_{u\in S_1} k(\mathbf{F}_u^{(K-1)}, \mathbf{F}_i^{(K-1)})\mathbf{F}_i^{(K-1)}$$

$$= \mathbf{X}_i + \frac{1}{|\mathcal{N}_i|}\left(\sum_{v_u\in\mathcal{N}_i\cap S_0}\mathbf{F}_u^{(K-1)} + \sum_{u\in\mathcal{N}_i\cap S_1}\mathbf{F}_u^{(K-1)}\right) + \frac{1}{N_0 N_1}\sum_{u\in S_1} k(\mathbf{F}_u^{(K-1)}, \mathbf{F}_i^{(K-1)})\mathbf{F}_u^{(K-1)}$$

$$- \frac{1}{N_0^2}\sum_{u\in S_0} k(\mathbf{F}_u^{(K-1)}, \mathbf{F}_i^{(K-1)})\mathbf{F}_u^{(K-1)} + (\frac{1}{N_0^2}\sum_{u\in S_0} k(\mathbf{F}_u^{(K-1)}, \mathbf{F}_i^{(K-1)}) - \frac{1}{N_0 N_1}\sum_{u\in S_1} k(\mathbf{F}_u^{(K-1)}, \mathbf{F}_i^{(K-1)}))\mathbf{F}_i^{(K-1)} \tag{39}$$

For the node $u \in \mathcal{S}_0$, a vector $\mathbf{F}_u^{(K)}$ satisfies $\mu_K^{(0)} - \Delta^K \preccurlyeq \mathbf{F}_u^{(K)} \preccurlyeq \mu_K^{(0)} + \Delta^K$, where $\mathbf{F}_u^{(K)} \preccurlyeq \mu_K^{(0)} + \Delta^K$ means that $\forall m \in \{1, 2, ..., D\}, F_{u,m} \le \mu_{K,m}^{(0)} + \Delta_m^K$. We abbreviate this as $\mathbf{F}_u^{(K)} \in \left[\mu_K^{(0)} \pm \Delta^K\right]$. Similarly, for the node $u \in \mathcal{S}_1$, $\mathbf{F}_u^{(K)} \in \left[\mu_K^{(1)} \pm \Delta^K\right]$. And we denote $\mu_K^{(0)}$ and $\mu_K^{(1)}$ as $\mu^{(0)}$ and $\mu^{(1)}$ with $K = 2$. Considering the unilateral case $v_i \in S_0$ and the above inequalities, we

have:

$$
\begin{aligned}
\mathbf{F}_i^{(K)} \in \Big[ & \mu^{(0)} + (\frac{|\mathcal{N}_i \cap \mathcal{S}_0|}{|\mathcal{N}_i|}\mu_{K-1}^{(0)} + \frac{|\mathcal{N}_i \cap \mathcal{S}_1|}{|\mathcal{N}_i|}\mu_{K-1}^{(1)}) - \frac{1}{N_0^2}\sum_{u \in S_0} k(\mathbf{F}_u^{(K-1)}, \mathbf{F}_i^{K-1})\mathbf{F}_u^{(K-1)} \\
& + \frac{1}{N_0 N_1}\sum_{u \in S_1} k(\mathbf{F}_u^{(K-1)}, \mathbf{F}_i^{(K-1)})\mu_{K-1}^{(1)} + \frac{1}{N_0^2}\sum_{u \in S_0} k(\mathbf{F}_u^{(K-1)}, \mathbf{F}_i^{(K-1)})\mathbf{F}_i^{(K-1)} \\
& - \frac{1}{N_0 N_1}\sum_{u \in S_1} k(\mathbf{F}_u^{(K-1)}, \mathbf{F}_i^{(K-1)})\mu_{K-1}^{(0)} \pm ((1 + \frac{2}{N_0})\boldsymbol{\Delta}^{K-1} + \boldsymbol{\Delta}) \Big] \\
\in \Big[ & \mu^{(0)} + \left(\mu_{K-1}^{(0)} + \frac{|\mathcal{N}_i \cap \mathcal{S}_1|}{|\mathcal{N}_i|}\left(\mu_{K-1}^{(1)} - \mu_{K-1}^{(0)}\right)\right) + \frac{1}{N_0 N_1}\sum_{u \in S_1} k(\mathbf{F}_u^{(K-1)}, \mathbf{F}_i^{(K-1)})(\mu_{K-1}^{(1)} - \mu_{K-1}^{(0)}) \\
& - \frac{1}{N_0^2}\sum_{u \in S_0} k(\mathbf{F}_u^{(K-1)}, \mathbf{F}_i^{(K-1)})\mathbf{F}_u^{(K-1)} + \frac{1}{N_0^2}\sum_{u \in S_0} k(\mathbf{F}_u^{(K-1)}, \mathbf{F}_i^{(K-1)})\mathbf{F}_i^{(K-1)} \pm ((1 + \frac{2}{N_0})\boldsymbol{\Delta}^{K-1} + \boldsymbol{\Delta}) \Big]
\end{aligned}
\tag{40}
$$

Let $\beta_i = |\mathcal{S}_{\text{opp}(v_i)}|/|\mathcal{N}_i|$ where $\mathcal{S}_{\text{opp}(v_i)}$ is the opposite sensitive group $\mathcal{N}_i \cap \mathcal{S}_1$ that $v_i$ belongs. For nodes in $S_0$, from Eq.(40), we have

$$
\begin{aligned}
\frac{1}{N_0}\sum_{i \in \mathcal{S}_0} \mathbf{F}_i^{(K)} \in \Big[ & \mu^{(0)} + \left(\mu_{K-1}^{(0)} + \frac{1}{N_0}\sum_{i \in \mathcal{S}_0} \beta_i(\mu_{K-1}^{(1)} - \mu_{K-1}^{(0)})\right) \\
& + \frac{1}{N_0^2 N_1}\sum_{i \in \mathcal{S}_0}\sum_{u \in \mathcal{S}_1} k(\mathbf{F}_u^{(K-1)}, \mathbf{F}_i^{(K-1)})(\mu_{K-1}^{(1)} - \mu_{K-1}^{(0)}) \pm \left((1 + \frac{2}{N_0})\boldsymbol{\Delta}^{K-1} + \boldsymbol{\Delta}\right) \Big]
\end{aligned}
\tag{41}
$$

For nodes in $\mathcal{S}_1$:

$$
\begin{aligned}
\frac{1}{N_1}\sum_{j \in \mathcal{S}_1} \mathbf{F}_j^{(K)} \in \Big[ & \mu^{(1)} + \left(\mu_{K-1}^{(1)} + \frac{1}{N_1}\sum_{j \in \mathcal{S}_1} \beta_j \left(\mu_{K-1}^{(0)} - \mu_{K-1}^{(1)}\right)\right) \\
& + \frac{1}{N_0 N_1^2}\sum_{j \in \mathcal{S}_1}\sum_{u \in \mathcal{S}_0} k(\mathbf{F}_u^{(K-1)}, \mathbf{F}_j^{(K-1)})(\mu_{K-1}^{(0)} - \mu_{K-1}^{(1)}) \pm ((1 + \frac{2}{N_1})\boldsymbol{\Delta}^{K-1} + \boldsymbol{\Delta}) \Big]
\end{aligned}
\tag{42}
$$

The fairness objective can be written as:

$$
\begin{aligned}
& \frac{1}{N_0}\sum_{i \in \mathcal{S}_0} \mathbf{F}_i^{(K)} - \frac{1}{N_1}\sum_{j \in \mathcal{S}_1} \mathbf{F}_j^{(K)} \in \\
& \Big[ (1 - (\frac{1}{N_0}\sum_{i \in \mathcal{S}_0} \beta_i + \frac{1}{N_1}\sum_{j \in \mathcal{S}_1} \beta_j) - (\frac{1}{N_0 N_1^2} + \frac{1}{N_0^2 N_1})\sum_{i \in \mathcal{S}_0}\sum_{j \in \mathcal{S}_1} k(\mathbf{F}_i^{(K-1)}, \mathbf{F}_j^{(K-1)}))(\mu_{K-1}^{(0)} - \mu_{K-1}^{(1)}) \\
& + (\mu^{(0)} - \mu^{(1)}) \pm ((2 + \frac{2}{N_0} + \frac{2}{N_1})\boldsymbol{\Delta}^{K-1} + 2\boldsymbol{\Delta}) \Big]
\end{aligned}
\tag{43}
$$

To obtain the upper bound for $\Delta_{\text{DP}}^{\text{Rep}}(\mathbf{F}^{(K)}) = \left\| \frac{1}{N_0} \sum_{i \in \mathcal{S}_0} \mathbf{F}_i^{(K)} - \frac{1}{N_1} \sum_{j \in \mathcal{S}_1} \mathbf{F}_j^{(K)} \right\|$, we have the following inequality:

$$
\begin{aligned}
\Delta_{\text{DP}}^{\text{Rep}}(\mathbf{F}^{(K)}) &= \left\| \frac{1}{N_0} \sum_{i \in \mathcal{S}_0} \mathbf{F}_i^{(K)} - \frac{1}{N_1} \sum_{j \in \mathcal{S}_1} \mathbf{F}_j^{(K)} \right\| \\
&\le |1 - (\frac{1}{N_0} \sum_{i \in \mathcal{S}_0} \beta_i + \frac{1}{N_1} \sum_{j \in \mathcal{S}_1} \beta_j) - (\frac{1}{N_0 N_1^2} + \frac{1}{N_0^2 N_1}) \sum_{i \in S_0} \sum_{j \in S_1} k(\mathbf{F}_i^{(K-1)}, \mathbf{F}_j^{(K-1)})| \left\| (\mu_{K-1}^{(0)} - \mu_{K-1}^{(1)}) \right\| \\
&\quad + \left\| \mu^{(0)} - \mu^{(1)} \right\| + ((2 + \frac{2}{N_0} + \frac{2}{N_1}) \left\| \mathbf{\Delta}^{K-1} \right\| + 2 \left\| \mathbf{\Delta} \right\|) \\
&\le (\frac{1}{N_0} \sum_{i \in \mathcal{S}_0} \beta_i + \frac{1}{N_1} \sum_{j \in \mathcal{S}_1} \beta_j) + 1 - (\frac{1}{N_0 N_1^2} + \frac{1}{N_0^2 N_1}) \sum_{i \in S_0} \sum_{j \in S_1} k(\mathbf{F}_i^{(K-1)}, \mathbf{F}_j^{(K-1)}) \left\| (\mu_{K-1}^{(0)} - \mu_{K-1}^{(1)}) \right\| \\
&\quad + \left\| \mu^{(0)} - \mu^{(1)} \right\| + ((2 + \frac{2}{N_0} + \frac{2}{N_1}) \left\| \mathbf{\Delta}^{K-1} \right\| + 2 \left\| \mathbf{\Delta} \right\|) \\
&\le (3 - (\frac{1}{N_0 N_1^2} + \frac{1}{N_0^2 N_1}) \sum_{i \in S_0} \sum_{j \in S_1} k(\mathbf{F}_i^{(K-1)}, \mathbf{F}_j^{(K-1)})) \left\| (\mu_{K-1}^{(0)} - \mu_{K-1}^{(1)}) \right\| \\
&\quad + \left\| \mu^{(0)} - \mu^{(1)} \right\| + ((2 + \frac{2}{N_0} + \frac{2}{N_1}) \left\| \mathbf{\Delta}^{K-1} \right\| + 2 \left\| \mathbf{\Delta} \right\|)
\end{aligned}
\tag{44}
$$

The upper bound of the fairness metric based on the Eq.(44) can be written as:

$$
\begin{aligned}
\Delta_{\text{DP}}^{\text{Rep}}(\mathbf{F}^{(K)}) &\le (3 - (\frac{1}{N_0 N_1^2} + \frac{1}{N_0^2 N_1}) \sum_{i \in S_0} \sum_{j \in S_1} k(\mathbf{F}_i^{(K-1)}, \mathbf{F}_j^{(K-1)})) \left\| (\mu_{K-1}^{(0)} - \mu_{K-1}^{(1)}) \right\| \\
&\quad + \left\| \mu^{(0)} - \mu^{(1)} \right\| + ((2 + \frac{2}{N_0} + \frac{2}{N_1}) \left\| \mathbf{\Delta}^{K-1} \right\| + 2 \left\| \mathbf{\Delta} \right\|).
\end{aligned}
\tag{45}
$$

In this paper, we consider the case with two-layer GNNs with $K = 2$, where $\mu_{K-2}^{(0)} - \mu_{K-2}^{(1)} = \mu^{(0)} - \mu^{(1)}$, $\mathbf{\Delta}_{K-2} = \mathbf{\Delta}$, $\mathbf{F}^{(K-2)} = \mathbf{X}$. We denote $3 - (\frac{1}{N_0 N_1^2} + \frac{1}{N_0^2 N_1}) \sum_{i \in S_0} \sum_{j \in S_1} k(\mathbf{F}_i^{(K-1)}, \mathbf{F}_j^{(K-1)})$ as $U$, where $U \le 3 - 1/N_0 - 1/N_1$. Based on the Eq.(35) and the previous equation, we can get:

$$
\begin{aligned}
U \left\| \mu_{K-1}^{(0)} - \mu_{K-1}^{(1)} \right\| &\le U \left( \left( 3 - (\frac{1}{N_0 N_1^2} + \frac{1}{N_0^2 N_1}) \sum_{i \in S_0} \sum_{j \in S_1} k(\mathbf{X}_i, \mathbf{X}_j) \right) \left\| (\mu^{(0)} - \mu^{(1)}) \right\| \right. \\
&\quad \left. + \left\| \mu^{(0)} - \mu^{(1)} \right\| + (2 + \frac{2}{N_0} + \frac{2}{N_1}) \left\| \mathbf{\Delta} \right\| + 2 \left\| \mathbf{\Delta} \right\| \right) \\
&\le U \left( (3 - \frac{1}{N_0} - \frac{1}{N_1}) \left\| (\mu^{(0)} - \mu^{(1)}) \right\| + (4 + \frac{2}{N_0} + \frac{2}{N_1}) \left\| \mathbf{\Delta} \right\| \right), \\
(2 + \frac{2}{N_0} + \frac{2}{N_1}) \| \mathbf{\Delta}^{K-1} \| &\le (2 + \frac{2}{N_0} + \frac{2}{N_1}) \left( (2 + \frac{4}{N_0} + \frac{4}{N_1}) \| \mathbf{\Delta} \| + 2 \| \mathbf{\Delta} \| \right) \\
&= 8(1 + \frac{1}{N_0} + \frac{1}{N_1})^2 \| \mathbf{\Delta} \|
\end{aligned}
\tag{46}
$$

We can finally get the upper bound of $\Delta_{\text{DP}}^{\text{Rep}}(\mathbf{F}^{(K)})$:

$$
\Delta_{\text{DP}}^{\text{Rep}}(\mathbf{F}^{(K)}) \le (3 - (\frac{1}{N_0 N_1^2} + \frac{1}{N_0^2 N_1}) \sum_{i \in S_0} \sum_{j \in S_1} k(\mathbf{F}_i^{(K-1)}, \mathbf{F}_j^{(K-1)})) C_3 + C_2,
\tag{47}
$$

where $C_2 = \left\| \mu^{(0)} - \mu^{(1)} \right\| + 8(1 + \frac{1}{N_0} + \frac{1}{N_1})^2 \| \mathbf{\Delta} \| + 2 \| \mathbf{\Delta} \|$. Also, the constant value $C_3$ is $C_3 = (3 - \frac{1}{N_0} - \frac{1}{N_1}) \left\| (\mu^{(0)} - \mu^{(1)}) \right\| + (4 + \frac{2}{N_0} + \frac{2}{N_1}) \| \mathbf{\Delta} \|$. $\qquad \square$

### G.3 EXTENSION OF THEOREM 4.2 TO $K > 2$

In this section, we discuss about the extension of the theorem 4.2 to $K > 2$. Here we use an example $K = 3$ and we can use similar ideas to get the theoretical results for other $K > 2$.

From Eq.(45), we can obtain:

$$
\begin{aligned}
\Delta_{\text{DP}}^{\text{Rep}}(\mathbf{F}^{(K)}) \leq & (3 - (\frac{1}{N_0 N_1^2} + \frac{1}{N_0^2 N_1}) \sum_{i \in S_0} \sum_{j \in S_1} k(\mathbf{F}_i^{(K-1)}, \mathbf{F}_j^{(K-1)})) \left\| (\mu_{K-1}^{(0)} - \mu_{K-1}^{(1)}) \right\| \\
& + \left\| \mu^{(0)} - \mu^{(1)} \right\| + ((2 + \frac{2}{N_0} + \frac{2}{N_1}) \left\| \mathbf{\Delta}^{K-1} \right\| + 2 \left\| \mathbf{\Delta} \right\|).
\end{aligned}
\tag{48}
$$

We consider the case with two-layer GNNs with $K = 3$, where $\mu_{K-3}^{(0)} - \mu_{K-3}^{(1)} = \mu^{(0)} - \mu^{(1)}$, $\mathbf{\Delta}_{K-3} = \mathbf{\Delta}$, $\mathbf{F}^{(K-3)} = \mathbf{X}$. We denote $3 - (\frac{1}{N_0 N_1^2} + \frac{1}{N_0^2 N_1}) \sum_{i \in S_0} \sum_{j \in S_1} k(\mathbf{F}_i^{(K-1)}, \mathbf{F}_j^{(K-1)})$ as $U$, where $U \leq 3 - 1/N_0 - 1/N_1$. Following the proof for Eq.(47), we can obtain:

$$
\Delta_{\text{DP}}^{\text{Rep}}(\mathbf{F}^{(K)}) \leq (3 - (\frac{1}{N_0 N_1^2} + \frac{1}{N_0^2 N_1}) \sum_{i \in S_0} \sum_{j \in S_1} k(\mathbf{F}_i^{(K-1)}, \mathbf{F}_j^{(K-1)})) C_3 + C_2,
\tag{49}
$$

where $C_2 = \left\| \mu_{K-2}^{(0)} - \mu_{K-2}^{(1)} \right\| + 8(1 + \frac{1}{N_0} + \frac{1}{N_1})^2 \| \mathbf{\Delta}^{K-2} \| + 2 \| \mathbf{\Delta}^{K-2} \|$ and $C_3 = (3 - \frac{1}{N_0} - \frac{1}{N_1}) \left\| (\mu_{K-2}^{(0)} - \mu_{K-2}^{(1)}) \right\| + (4 + \frac{2}{N_0} + \frac{2}{N_1}) \| \mathbf{\Delta}^{K-2} \|$. Following the proof in Eq.(46), we have:

$$
\begin{aligned}
\left\| \mu_{K-2}^{(0)} - \mu_{K-2}^{(1)} \right\| & \leq (3 - \frac{1}{N_0} - \frac{1}{N_1}) \left\| (\mu^{(0)} - \mu^{(1)}) \right\| + (4 + \frac{2}{N_0} + \frac{2}{N_1}) \| \mathbf{\Delta} \|, \\
\| \mathbf{\Delta}^{K-2} \| & \leq 4(1 + \frac{1}{N_0} + \frac{1}{N_1}) \| \mathbf{\Delta} \|.
\end{aligned}
\tag{50}
$$

Therefore, we can obtain the following upper bound for $K = 3$:

$$
\Delta_{\text{DP}}^{\text{Rep}}(\mathbf{F}^{(K)}) \leq (3 - (\frac{1}{N_0 N_1^2} + \frac{1}{N_0^2 N_1}) \sum_{i \in S_0} \sum_{j \in S_1} k(\mathbf{F}_i^{(K-1)}, \mathbf{F}_j^{(K-1)})) C_5 + C_4,
\tag{51}
$$

where $C_4 = (3 - \frac{1}{N_0} - \frac{1}{N_1}) \left\| (\mu^{(0)} - \mu^{(1)}) \right\| + (8 + \frac{6}{N_0} + \frac{6}{N_1}) \| \mathbf{\Delta} \| + 32(1 + \frac{1}{N_0} + \frac{1}{N_1})^3 \| \mathbf{\Delta} \|$ and $C_5 = (3 - \frac{1}{N_0} - \frac{1}{N_1})^2 \left\| (\mu^{(0)} - \mu^{(1)}) \right\| + (3 - \frac{1}{N_0} - \frac{1}{N_1})(4 + \frac{2}{N_0} + \frac{2}{N_1}) \| \mathbf{\Delta} \| + (4 + \frac{2}{N_0} + \frac{2}{N_1}) 4(1 + \frac{1}{N_0} + \frac{1}{N_1}) \| \mathbf{\Delta} \|$.

From the Eq.(51) with $K = 3$, we can observe that the upper bound of the fairness metric $\Delta_{\text{DP}}^{\text{Rep}}(\mathbf{F}^{(K)})$ is only relevant to the third term of MMD loss, where it has the same theoretical results as $K = 2$. We can use the idea above to get the theoretical results of other $K$ with $K > 2$ and their upper bound have similar format. Therefore, we can also have the same theoretical results as $K = 2$, which means that our theorem 4.2 can be used for all $K > 2$.

## H EXPERIMENTAL DETAILS

### H.1 DATASETS DETAILS AND STATISTICS

#### H.1.1 HOMOPHILY RATIO BASED ON LABELS

We utilize the following edge homophily ratio (Zhu et al., 2020) to measure the one-hop neighbor homophily of the graph. Specifically, the edge homophily ratio $\mathcal{H}(\mathcal{G})$ is the proportion of edges that connect two nodes of the same class:

$$
\mathcal{H}(\mathcal{G}) = \frac{|\{(u, v) : (u, v) \in \mathcal{E} \wedge y_u = y_v\}|}{|\mathcal{E}|}
\tag{52}
$$

Table 4: Statistics of used homophily and heterophily graph datasets in this paper.

| Dataset | #Nodes | # Edges | #Features | $\mathcal{H}(\mathcal{G})$ | $\mathcal{H}_f(\mathcal{G})$ | Sens | Label |
|---------|--------|---------|-----------|------|--------|------|-------|
| German | 1,000 | 22,242 | 27 | 0.59 | 0.81 | Gender | Good/bad Credi |
| Credit | 3,327 | 4,552 | 13 | 0.74 | 0.97 | Age | Default/no default Payment |
| Bail | 19,717 | 44,324 | 18 | 0.81 | 0.52 | Race | Bail/no bail |

### H.1.2 SENSITIVE ATTRIBUTE HOMOPHILY

We utilize the following sensitive attribute homophily ratio (Jiang et al., 2023) to measure the one-hop neighbor homophily of the graph based on sensitive attributes. Given a graph $\mathcal{G} = (\mathcal{V}, \mathcal{E}, \mathbf{X})$ and sensitive attributes $\mathbf{s}$, the sensitive homophily ratio $\mathcal{H}_f$ for $\mathcal{G}$ is defined as:

$$\mathcal{H}_f = \frac{|\{(u,v) : (u,v) \in \mathcal{E} \wedge s_u = s_v\}|}{|\mathcal{E}|}. \tag{53}$$

It measures the ratio of edges connecting nodes from the same sensitive attributes. Intuitively, GNNs may give biased predictions on graphs with more intra sensitive group edges (higher $\mathcal{H}_f$) (Dai & Wang, 2021; Dong et al., 2022; Jiang et al., 2023) because more intra group edges may lead to similar representations of nodes from the same sensitive groups, resulting in biased predictions, i.e., predictions highly correlated with sensitive attributes.

### H.2 DATASETS

The statistics of datasets, including their homophily ratio based on labels and sensitive homophily ratio, are given in Table 4. We can observe all graphs exhibit higher sensitive homophily ratio so original message passing on the graph structure can exacerbate the fairness issues on graphs.

**German Credit (German)** (Agarwal et al., 2021): In this graph, nodes represent clients in a German bank and attributes of nodes are gender, loan amount, and other account related details. Edges between two nodes (clients) mean their credit accounts are similar. The label is weather the credit risk of the clients as high or low. The attribute "gender" is treated as a sensitive feature.

**Recidivism (Bail)** (Agarwal et al., 2021): In this graph, defendants released on bail during 1990-2009 are represented as nodes. If two defendants share similar past criminal records and demographics, an edge is added between these two nodes. The task is to predict whether a defendant would be more likely to commit a violent or nonviolent crime once released on bail. The attribute "race" is the sensitive feature.

**Credit Defaulter (Credit)** (Agarwal et al., 2021): Nodes in this dataset represent credit card users and edges indicate if two users share a similar pattern in purchases/payments. The class labels of nodes represent whether a user will default on credit card payment with "age" being the sensitive feature.

The statistics of datasets are given in Table 4. We follow the split introduced in the previous papers for graph group fairness (Agarwal et al., 2021; Dong et al., 2022).

### H.3 BASELINES

**GCN** (Kipf & Welling, 2016): GCN is one of the most popular spectral GNN models based on graph Laplacian, which has shown great performance for node classification.

**GIN** (Xu et al., 2018): GIN proposed a powerful GNN by approximating injective set functions with multi-layer perceptrons (MLPs), which can be as powerful as the 1-WL test.

**GAT** (Veličković et al., 2017): Instead of treating each neighbor nodes equally, GAT utilizes an attention mechanism to assign different weights to nodes in the neighborhood during the aggregation step.

Table 5: Model performance with GIN and GAT as backbones. The best and second best performance are marked with **boldface** and underline.

| Dataset | Metric | GIN | | | | GAT | | | |
|---|---|---|---|---|---|---|---|---|---|
| | | Vanilla | FairVGNN | GMMD | GMMD-S | Vanilla | FairVGNN | GMMD | GMMD-S |
| German | AUC (↑) | **72.71**±1.44 | 71.65±1.90 | 71.42±0.80 | 71.30±1.24 | **70.95**±2.34 | 68.59±3.80 | 70.32±0.20 | 70.10±2.08 |
| | F1 (↑) | 82.46±0.89 | 82.14±1.40 | 82.11±0.35 | **82.48**±0.18 | 81.18±0.69 | 82.07±0.55 | **82.57**±0.13 | 82.43±0.32 |
| | ACC (↑) | **73.44**±1.09 | 70.16±0.32 | 70.27±0.38 | 70.53±0.75 | 68.32±2.10 | 69.76±0.48 | **70.67**±0.19 | 70.27±0.68 |
| | $\Delta_{DP}$ (↓) | 35.17±7.27 | 0.43±0.53 | 0.18±0.26 | **0.09**±0.13 | 4.68±4.27 | 1.83±3.66 | **0.39**±0.26 | 0.60±0.51 |
| | $\Delta_{EO}$ (↓) | 25.17±5.89 | 0.34±0.41 | 0.74±1.05 | **0.26**±0.37 | 4.35±3.28 | 1.64±3.28 | 1.02±0.36 | **0.51**±0.36 |
| Credit | AUC (↑) | **74.36**±0.21 | 71.36±0.72 | 70.63±0.67 | 70.35±0.53 | **72.94**±1.84 | 71.68±3.57 | 70.52±0.15 | 69.26±0.05 |
| | F1 (↑) | 82.28±0.64 | 87.44±0.23 | **87.60**±0.04 | 86.23±0.75 | 85.43±0.63 | 85.04±2.09 | **85.83**±2.52 | 85.43±0.54 |
| | ACC (↑) | 74.02±0.73 | 78.18±0.20 | **78.23**±0.45 | 77.62±0.82 | **77.31**±0.54 | 76.41±1.45 | 76.82±2.04 | 76.75±0.43 |
| | $\Delta_{DP}$ (↓) | 14.48±1.24 | 2.85±2.10 | **0.12**±0.12 | 2.69±1.13 | 19.42±4.52 | 7.13±6.58 | **1.38**±1.32 | 1.67±0.42 |
| | $\Delta_{EO}$ (↓) | 12.35±2.68 | 1.72±1.80 | **0.16**±0.16 | 1.99±1.00 | 15.79±5.00 | 5.90±5.82 | **1.32**±1.27 | 3.08±1.10 |
| Bail | AUC (↑) | 86.14±0.25 | 83.22±1.60 | **86.53**±1.54 | 86.50±1.61 | 88.10±3.52 | 87.46±0.69 | **90.58**±0.74 | 89.77±0.74 |
| | F1 (↑) | 76.49±0.57 | 76.36±2.20 | 76.62±1.88 | **76.64**±1.71 | 76.80±5.54 | 76.12±0.92 | **78.45**±3.21 | 77.34±0.47 |
| | ACC (↑) | 81.70±0.67 | 83.86±1.57 | **84.08**±1.35 | 84.04±1.26 | **83.71**±4.55 | 81.56±0.94 | 83.39±4.50 | 83.44±2.13 |
| | $\Delta_{DP}$ (↓) | 8.55±1.61 | 5.67±0.76 | **3.57**±1.11 | 3.61±1.42 | 5.66±1.78 | 5.41±3.25 | **1.24**±0.28 | 3.27±1.79 |
| | $\Delta_{EO}$ (↓) | 6.99±1.51 | 5.77±1.26 | **4.69**±0.69 | 4.71±0.66 | 4.34±1.35 | 2.25±1.61 | **1.18**±0.34 | 2.52±2.22 |

**NIFTY** (Agarwal et al., 2021): NIFTY has been introduced to enhance counterfactual fairness and stability of node representations. This involves a novel triplet-based objective function and layer-wise weight normalization using the Lipschitz constant.

**EDITS** (Dong et al., 2022): EDITS employs the Wasserstein distance approximator to alternately debias node features and network topology

**FairGNN** (Dai & Wang, 2021): FairGNN uses a sensitive feature estimator to increase the quantity of sensitive attributes, which is highly effective in promoting fairness within their algorithm.

**FairVGNN** (Wang et al., 2022): FairVGNN automatically masks feature channels that are correlated with sensitive attributes and adaptively adjusts the encoder weights to absorb less sensitive information during feature propagation, leading to a more fair and unbiased machine learning algorithm.

**FMP** (Jiang et al., 2022): FMP uses a unified optimization framework to develop an effective message passing to learn fair node representations while preserving prediction performance.

**MMD** (Gretton et al., 2006): MMD is about measuring the discrepancy between two distribution and it can be used as fairness regularization to train fair predictors (Louizos et al., 2015). For this baseline, we immediately add this regularization at the prediction level of GNN models.

### H.4 Setup and Hyper-parameter settings

We use official implementation publicly released by the authors on Github of the baselines or implementation from Pytorch Geometric. For fair comparison, we used grid search to find the best hyperparameters of the baselines. Note that we use two-layer GNN models for all methods. We run experiments on a machine with a NVIDIA Tesla V100 GPU with 32GB of GPU memory. In all experiments, we use the Adam optimizer (Kingma & Ba, 2014). A small grid search is used to select the best hyperparameter for all methods. In particular, for our GMMD. we search $\lambda_s$ from $\{0, 0.1, 0.5, 1, 2\}$, $\lambda_f$ from $\{0, 0.1, 0.5, 1, 5, 10\}$, the hidden dimension of $M$ layers MLP is fixed as 16, $\alpha$ from $\{0.1, 0.3, 0.7, 1\}$ and $M$ from $\{1, 2, 4, 6\}$. We tune the learning rate over $\{0.003, 0.001, 0.0001\}$ and weight decay over $\{0, 0.4, 0.0001, 0.00001\}$. We select the best configuration of hyper-parameters based on accuracy, f1 score, AUC, $\Delta_{DP}$ and $\Delta_{EO}$ only based on the validation set. The sampled number for German, Bail and Credit are 100, 200 and 6000.

## I Additional Experiments

### I.1 Additional Results of Different Backbones

In this section, we show the performance of models with different backbones and we select the state-of-the-art method FairVGNN as baselines to verify the effectiveness of our proposed model. These results also echo the conclusion in the main text: our model can achieve great performance with

regard to both utility metric and fairness metric compared with state-of-the-art method FairVGNN. It shows the flexibility of our model by adopting different message passing approaches.

## I.2 ADDITIONAL HYPER-PARAMETER ANALYSIS

In this section, we investigate the hyperparameters of $\alpha$ and the number of MLP layers $M$.

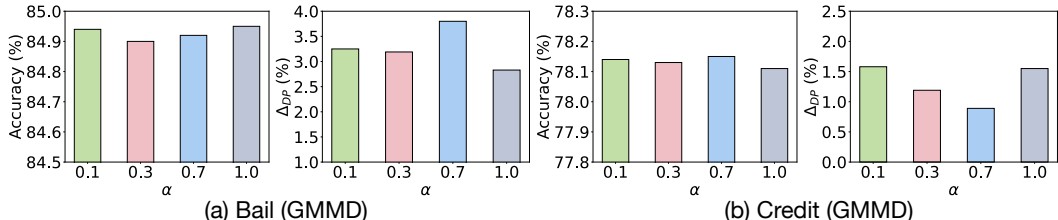

Figure 4: Hyperparameter Analysis of $\alpha$ on German and Credit.

**Hyperparameter Analysis of $\alpha$.** We provide the hyperparameter analysis of $\alpha$ for the kernel similarity calculation. $\alpha$ is varied as $\{0.1, 0.3, 0.7, 1\}$. Figure 4 shows the results on Bail and Credit. We can first observe that $\alpha$ is only used for the fairness regularization term so it doesn't have too much influence on the accuracy performance. Moreover, larger values of $\alpha$ (i.e. 0.7, 1) can promote the performance on the fairness metric. One possible reason is that larger $\alpha$ can also increase the weight for the fairness regularization in Eq.(7) and Eq.(4) to learn a more fair model.

**Hyperparameter Analysis of $M$.** We run a sweep over the number of layers for MLP $M$ to study how it affects the performance. We vary $M$ as $\{1, 2, 4, 6\}$. Figure 5 shows the corresponding results on Bail and Credit datasets. Larger number of $M$ represents deeper MLP layers and can first increase the performance on Accuracy and then decrease it. This is because deeper MLP can increase the expressive power of the initial representation for our message passing. However, too many layers will lead to a large number of parameters and can easily overfit the training data, which will achieve worse performance. Also, a small number of layers (i.e, $M = 2$ or $M = 4$) is enough to achieve good performance on the fairness metric.

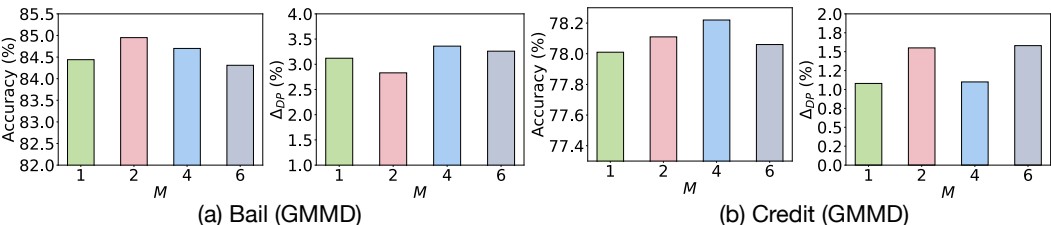

Figure 5: Hyperparameter Analysis of $M$ on German and Credit.

**Effect of Layers $K$.** We then explore the effect of layers $K$ on the utility (AUC, F1, Acc) and fairness ($\Delta_{\text{DP}}$ and $\Delta_{\text{EO}}$). Figure 6 shows the corresponding results. We can find that (**i**) the utility performance first increases and then decreases with the increase of $K$. This is because a larger $K$ leads to over-smoothing issues, which will degenerate the model's performance; and (**ii**) $\Delta_{DP}$ and $\Delta_{EO}$ consistently drop as $K$ increases. This is because more GMMD layers can effectively minimize the MMD loss and lead to a fairer model. To learn a fair and accurate model, the number of layers $K$ can be selected from 2 to 4 based on our experimental results.

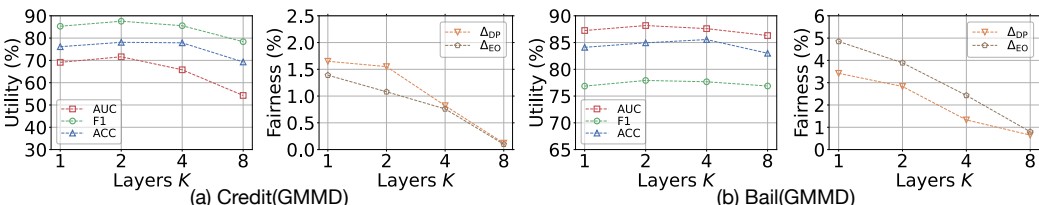

Figure 6: Different $K$ layers analysis on Bail and Credit datasets.

## I.3    ADDTIONAL RESULTS ON CASE STUDY

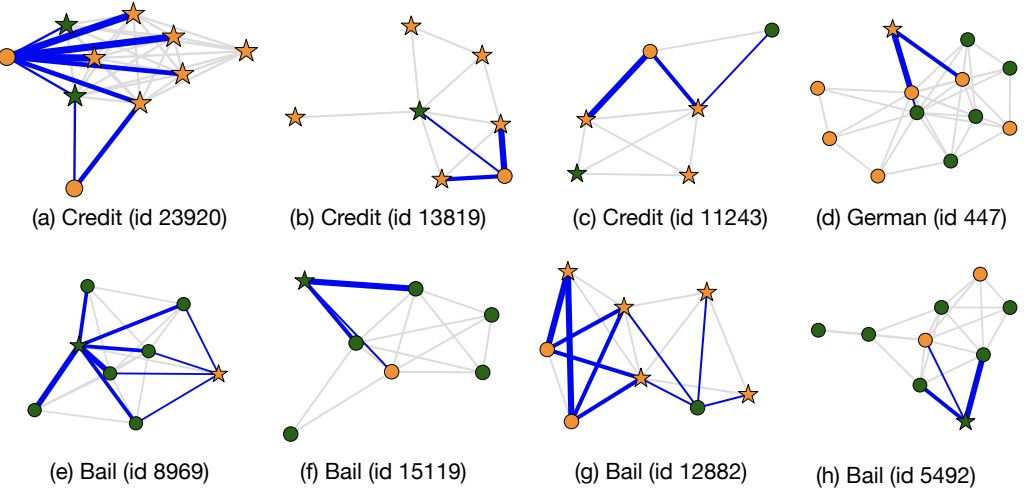

Figure 7: Additional Case Study on all three datasets.

In Figure  7, we provide the additional case studies on all three datasets. We can find that, in most cases, our model can successfully assign higher weights to inter sensitive group edges which connect two nodes from the same label. This observation interprets the reason why GMMD can also achieve good utility performance when aggregating nodes from different sensitive groups to mitigate fairness issues.

Table 6: Comparison of the running time where "s" indicates seconds.

| Dataset | NIFTY | FairGNN | EDITS | FairVGNN | FMP | GMMD | GMMD-S |
|---------|-------|---------|-------|----------|-----|------|--------|
| German  | 26.1s | 20.8s   | 25.7s | 104.3s   | 15.1s | 16.8s | 15.2s |
| Bail    | 27.5s | 20.1s   | 33.3s | 99.2s    | 19.2s | 21.3s | 19.8s |
| Credit  | 38.1s | 26.3s   | 260.5s | 92.4s   | 22.7s | 25.5s | 23.6s |

## I.4    RUNNING TIME COMPARISON

We also conduct experiments to compare our training time with baselines. For comparison, we consider five representative and state-of-the-art baselines that achieve great performance in mitigating fairness issues for node classification, including NIFTY, FairGNN and EDITS, FMP, GMMD and GMMD-S. For all methods, we use GCN as the backbone for running time calculation. We conduct all running time experiments on the same machine with a NVIDIA Tesla V100 GPU (32GB of GPU memory). Furthermore, we count the running time of the models with multiple epochs which achieve the best results. Table 6 shows the results of running time for all models. Note that we include the total training time for the two steps of the EDITS model in our table, reflecting the combined time required for both steps. Specifically, the first step of EDITS is to learn a new adjacency matrix that removes sensitive information related to the graph structure. The second step is to train a GNN model with this new adjacency matrix. The results show the efficiency of GMMD and GMMD-S, and we can find the superior performance of GMMD and GMMD-S does not benefit from a larger model complexity.

## J    ADDITIONAL DISCUSSION ON POTENTIAL LIMITATIONS AND FUTURE DIRECTIONS

In this paper, we proposed a novel message passing method called GMMD, along with an efficient and simplified version called GMMD-S to mitigate biased prediction results of GNNs. The theoretical analysis and our experiment results validate the effectiveness of our model.

Despite the theoretical analysis and great performance of our model, one limitation of current work and some interesting directions we want to explore are: our message passing method is derived from smoothness and fairness regularization, Maximum Mean Discrepancy (MMD), which can provide a better intuitive and theoretical understanding. However, there are various fairness regularization methods and it's also promising to design a new regularization term for learning fair representation of graph structure. Moreover, it's an interesting direction to conduct a theoretical analysis of the utility performance of the proposed fairness-aware message passing, which can help us have a better understanding of controlling the trade-off between utility and fairness.

