# OpenReview forum: "Fairness-aware Message Passing for Graph Neural Networks"
_ICLR.cc/2024/Conference — ICLR 2024 Conference Withdrawn Submission_

### Official Review · Reviewer_6j2s · 2023-10-29

**Soundness:** 3 good
**Presentation:** 3 good
**Contribution:** 2 fair
**Rating:** 5
**Confidence:** 4

**Summary:**

This works presents a fairness-aware GNN layer that is built on a theoretical bias analysis and provides a theoretical fairness guarantee for the proposed layer. The layer incorporates both smoothness and fairness regularizers providing a utility/fairness trade-off. The main motivation is to increase the impact of representations from different sensitive groups in the message passing in order to bring the representations from different sensitive groups closer.

**Strengths:**

- The paper's proposed solution for bias mitigation is efficient to implement and intuitive.
- The proposed fairness-aware GNN layer is the first one with theoretical fairness guarantees to the best of my knowledge.
- Experimental results are satisfactory and the considered baselines are sufficient. Furthermore, the visualization and ablation study are helpful in interpreting the presented results.
- The paper is fluent (except for a few typos).

**Weaknesses:**

- In terms of theoretical analyses and intuition, the novelty is limited.
- Generally, real-world networks are sparse in inter-edges (the datasets in the experiments are artifically created based on the similarities of nodal attributes). The proposed method might be limited in mitigating bias for the case of sparse inter-edges, as the number
 of neighboring nodes from different sensitive groups might be 0 or small for this case.
- For the baselines, the values over which the grid searches are run are not provided.

**Questions:**

I believe this work has certain similarities to some existing works. In terms of the utilized technique, it is highly similar to FMP. Furthermore, the overall scheme is basically equivalent to balancing inter-/intra-edges in message passing, which is already theoretically justified [1, 2] and implemented via optimization [1], or augmentations [2,3,4] before. However, the proposed fair GNN layer is the first one with a theoretical fairness guarantee and seems to be empirically well-performing. Although the intuition and analyses are not very novel, the implementation of these ideas is very systematic and efficient, which provides enough novelty for this work. However, I have the following suggestions to the Authors.

- For networks that are sparse in terms of inter edges (like Pokec networks [5]) the proposed method might not be as efficient as it is on denser networks. For this case, augmentation based inter-/intra-edge balancing techniques might be more advantageous, i.e., FairAug [2]. Further experimental results over Pokec networks can be useful to better understand this limitation.

- I do not agree with the claim that “FMP is still not effective and lacks a theoretical understanding of how its message passing process can mitigate biases in GNNs”. As the regularizer employed in FMP directly minimizes demographic parity, it is naturally clear and justified how it mitigates bias. I think the discussion in Appendix A regarding the differences of the proposed algorithm and FMP presents a better picture for novelties of current work compared to FMP. I suggest a summary of those novelties instead of the aforementioned claim to present a clearer literature review.

- Please provide the hyperparameter values over which the grid searches are run for the baselines.

- I have realized a few typos while reading the paper. I suggest another round of proof-reading .

[1] Li, Peizhao, et al. "On dyadic fairness: Exploring and mitigating bias in graph connections." International Conference on Learning Representations. 2020.

[2] Kose, O. Deniz, and Yanning Shen. "Demystifying and Mitigating Bias for Node Representation Learning." IEEE Transactions on Neural Networks and Learning Systems (2023).

[3] Spinelli, Indro, et al. "Fairdrop: Biased edge dropout for enhancing fairness in graph representation learning." IEEE Transactions on Artificial Intelligence 3.3 (2021): 344-354.

[4] Ling, Hongyi, et al. "Learning fair graph representations via automated data augmentations." The Eleventh International Conference on Learning Representations. 2022.

[5] Dai, Enyan, and Suhang Wang. "Say no to the discrimination: Learning fair graph neural networks with limited sensitive attribute information." Proceedings of the 14th ACM International Conference on Web Search and Data Mining. 2021.

---

### Official Review · Reviewer_Kxjw · 2023-10-30

**Soundness:** 3 good
**Presentation:** 3 good
**Contribution:** 2 fair
**Rating:** 5
**Confidence:** 4

**Summary:**

The paper introduces GMMD, a new fairness-aware message passing framework, tackling both graph smoothness and representation fairness. GMMD promotes node representation aggregation from different sensitive groups while minimizing representation from the same group, fostering fair representations. A theoretical analysis and a simpler variant, GMMD-S, are provided, demonstrating GMMD's capability in enhancing fairness across different GNN models without compromising accuracy.

**Strengths:**

- The paper advances research on fairness within GNNs through a message-passing lens, presenting a fresh perspective that enriches the discourse in this field.
- The paper furnishes a comprehensive theoretical analysis, adding a robust foundation to the proposed framework which enhances the overall rigor of the study.

**Weaknesses:**

My primary concern regarding this paper centers around its experimental performance. The absence of three notable datasets: Pokec-n, Pokec-c, and NBA, which are commonly used benchmarks in this domain. Most of the baseline methods, against which your method is compared, have been evaluated on these three datasets in their original papers. Consequently, without the inclusion and evaluation on these datasets, the performance claims made in this paper remain unconvincing in my view.

**Questions:**

I have no question.

---

### Official Review · Reviewer_ci7T · 2023-11-01

**Soundness:** 2 fair
**Presentation:** 3 good
**Contribution:** 2 fair
**Rating:** 3
**Confidence:** 4

**Summary:**

The paper propose a fairness-aware message passing framework GMMD to mitigate biased predictions on graph data, which guarantees graph smoothness and enhances fairness. The authors also show that the proposed message passing can achieve better downstream performance with regard to fairness metrics from both theoretical and empirical perspective.

**Strengths:**

1.This paper interprets message passing as modeling inter-sensitive-group and intra-sensitive-group relations to learn fair representation. Thus GMMD can be intuitively interpreted as encouraging a node to aggregate representations of other nodes from different sensitive groups while subtracting representations of other nodes from the same sensitive group.

2.Adequate experimental comparisons were made with multiple baselines, and the results demonstrate the effectiveness of the proposed method.

3.The theoretical analysis is solid.

**Weaknesses:**

1.Even with the simplification of MMD, the computational complexity of the resulting simplified optimization objective is still high due to the need for similarity computation. While the sampling strategy addresses this issue to some extent, the simplicity of the sampling strategy raises the question of whether random sampling could lead to unstable performance.

2.The paper aligns with the goals of existing work(EDITS[1],FMP[2]) in terms of reducing disparities in representations among different sensitive groups. The paper also tries to improving message passing, a problem shared with FMP. However, its innovation is limited to derived theoretical  analysis to improve the message passing from a slightly different perspective. And from both a methodological and motivational perspective, it lacks novelty.

**Questions:**

1. From the perspective of related work and motivation: although Appendix A explains the differences with FMP in detail, it seems to me that both are motivated to achieve fair message passing by reducing disparities in the representations between different sensitive groups in the final outcome. GMMD, similar to FMP and EDITS, achieves this through regularization terms to constrain the gradient updatings and the BP process.

2. From the perspective of contribution: from what I understand, the approach proposed in the paper aims to improve GCN's message passing by introducing regularization terms to reduce the disparities in representations among different sensitive groups. However, this approach may not be considered highly novel, as the idea of using regularization to achieve fairness in message passing is not entirely new. What sets this paper apart, as claimed by the authors, is the improvement made to the MMD technique and the theoretical proof that distinguishes it from FMP. In my opinion, the level of innovation is somewhat limited. Furthermore, because the method modifies the message passing, it has to change the GNN backbones. Therefore, it can not be considered a plug-and-play method and may have lower transferability compared to some other plug-and-play or data-centric approaches.

---

### Official Review · Reviewer_qojd · 2023-11-01

**Soundness:** 2 fair
**Presentation:** 3 good
**Contribution:** 2 fair
**Rating:** 5
**Confidence:** 4

**Summary:**

This paper proposes a novel graph neural network framework called GMMD (Graph Neural Network with Maximum Mean Discrepancy-based Fairness-aware Message Passing), which aims to address the shortcomings of the existing graph neural networks in facing the fairness problem. GMMD simultaneously maintains the model's high accuracy and group fairness on node classification tasks by introducing a Maximum Mean Discrepancy (MMD)-based fairness-aware message passing mechanism. In addition, the authors introduce a simplified version of GMMD, GMMD-S, which further improves the efficiency of the algorithm. The results on three real-world datasets are present.

**Strengths:**

1.	The paper presents GMMD, an innovative fairness-centric message passing framework specifically designed for Graph Neural Networks (GNNs). GMMD distinctively promotes fairness by encouraging nodes to integrate representations from diverse sensitive groups, while simultaneously minimizing the influence of representations from identical sensitive groups. This methodology uniquely tackles structural bias in graph learning, offering a fresh and effective perspective.

2.	The authors evaluated their approach across multiple real-world datasets, showcasing its practical applicability and reliability in diverse scenarios.

3.	The method proposed by the authors can be simply applied to various GNN model structures. This adaptability significantly broadens the method's applicability, allowing it to enhance the performance of GNN models across a diverse range of applications.

**Weaknesses:**

1.	While the authors provide a comprehensive theoretical analysis in the methods section, it remains unclear how the trade-off between model fairness and performance is handled. Specifically, the impact of increasing the weight of neighboring nodes with differing sensitive attributes on node embedding needs to be further elaborated. Does this approach result in an overemphasis on neighboring nodes with different sensitive attributes?


2.	The paper could benefit from an expanded discussion on the role of neighboring nodes, particularly those within a few hops of the central node in an ego-graph. These nodes, which typically share the same sensitive attributes as the central node, play a significant role in influencing the central node. The authors should explore whether reducing the influence of these core neighbors compromises the quality of node embeddings, potentially leading to a less realistic representation space,  i.e., the graph reconstruction loss increases.

3.	The paper needs additional clarification on certain notation definitions, particularly P_ij in Eq. 3. The authors should specify whether N is derived from the input graph or the ego-graph when calculating the number of samples. Furthermore, given the tendency for data distribution in graphs to be unbalanced, it will benefit if the authors  discuss the potential impact on minority and majority sensitivity groups. If there is a substantial size disparity between these groups, does the model become overly sensitive to minority classes while underserving majority classes?

**Questions:**

The paper mentions the significant influence of neighboring nodes within an ego-graph, especially those a few hops away from the central node that share the same sensitive attributes. I appreciate if the authors could delve deeper into whether diminishing the impact of these core neighbors could degrade the quality of the node embeddings, and thus result in a less authentic representation space, as indicated by an increase in graph reconstruction loss? Additionally, could you provide more clarity on the notation P_ij in Eq. 3, particularly regarding whether N is calculated based on the input graph or the ego-graph? Lastly, how does the model account for potential imbalances in the distribution of data across sensitivity groups, ensuring it does not become overly responsive to minority classes at the expense of majority classes?

---

### Official Review · Reviewer_UpE8 · 2023-11-06

**Soundness:** 2 fair
**Presentation:** 2 fair
**Contribution:** 2 fair
**Rating:** 5
**Confidence:** 3

**Summary:**

The paper considers the goals of achieving both graph smoothness and fairness in message-passing at the same time. A framework GMMD, as well as the simpler version GMMD-S, is proposed. Theoretical derivations and empirical experiments are also presented.

**Strengths:**

The strength of the paper comes from the emphasis on the two goals related to fairness in the context of GNNs. It is nice to see the effort to derive new approaches for message passing and are fairness-aware (as noted by authors).

**Weaknesses:**

The paper is not easy to follow from time to time. For example, the second to the last paragraph in Section 1, the sentence "Intuitively ..." spans multiple lines, packed densely with intuitions and motivations. As another example, what does the proposed framework GMMD stand for (there does not seem to be a full name presented in the manuscript)? It would be helpful if authors can correct the typos (some of them are listed below in __Questions__), rearrange materials (especially in Section 4) to present more clearly the takeaway messages along with the theoretical derivations.

**Questions:**

__Q1__: w.r.t. the notion of fairness of interest and the motivation behind GMMD

Considering that there are multiple group-level fairness notions proposed in the algorithmic fairness literature, it would be clearer if the notion of interest can be explicitly noted early on in the paper. The MMD metric utilized in the proposed GMMD framework is for the purpose of quantifying group-wise distribution discrepancies, which indicates that the applicability of GMMD is limited to DP notion of fairness. However, the first appearance of DP notion is in Section 4.2, after introducing the GMMD framework. I am wondering if GMMD is general enough to handle various fairness notions, or DP is the actual motivation behind GMMD framework.

__Q2__: applicability of the framework and theoretical results

The theoretical results are derived with binary classification and binary sensitive features, and with $K = 2$. The values of the involved constants also seem rather specifically tailored to the aforementioned setting. Additional discussions and clarifications on the takeaway message and the applicability of results would be very helpful.


__Minor typos:__

- top of page 2 (first line): "fairness-ware"

- bottom of page 2 (last but second line): "fairness-ware"

- Section 4.1 (first paragraph): "on step gradient"